# Loci for human leukocyte telomere length in the Singaporean Chinese population and trans-ethnic genetic studies

Rajkumar Dorajoo [1,14], Xuling Chang[2,3,14], Resham Lal Gurung[4,14], Zheng Li [1], Ling Wang[1], Renwei Wang[5], Kenneth B. Beckman [6], Jennifer Adams-Haduch[5], Yiamunaa M[4], Sylvia Liu[4], Wee Yang Meah[1], Kar Seng Sim [1], Su Chi Lim[4,7,8], Yechiel Friedlander[9], Jianjun Liu [1,10], Rob M. van Dam[8,10], Jian-Min Yuan [5,11,15], Woon-Puay Koh[8,12,15], Chiea Chuen Khor [1,13,15] & Chew-Kiat Heng [2,3,15]

Genetic factors underlying leukocyte telomere length (LTL) may provide insights into telomere homeostasis, with direct links to disease susceptibility. Genetic evaluation of 23,096 Singaporean Chinese samples identifies 10 genome-wide loci ($P < 5 \times 10^{-8}$). Several of these contain candidate genes (*TINF2*, *PARP1*, *TERF1*, *ATM* and *POT1*) with potential roles in telomere biology and DNA repair mechanisms. Meta-analyses with additional 37,505 European individuals reveals six more genome-wide loci, including associations at *MPHOSPH6*, *NKX2-3* and *TYMS*. We demonstrate that longer LTL associates with protection against respiratory disease mortality [HR = 0.854(0.804–0.906), $P = 1.88 \times 10^{-7}$] in the Singaporean Chinese samples. We further show that the LTL reducing SNP rs7253490 associates with respiratory infections ($P = 7.44 \times 10^{-4}$) although this effect may not be strongly mediated through LTL. Our data expands on the genetic basis of LTL and may indicate on a potential role of LTL in immune competence.

[1] Genome Institute of Singapore, Agency for Science, Technology and Research, Singapore 138672, Singapore. [2] Department of Paediatrics, Yong Loo Lin School of Medicine, National University of Singapore, Singapore 119228, Singapore. [3] Khoo Teck Puat – National University Children's Medical Institute, National University Health System, Singapore 119074, Singapore. [4] Clinical Research Unit, Khoo Teck Puat Hospital, Singapore 768828, Singapore. [5] Division of Cancer Control and Population Sciences, UPMC Hillman Cancer Center, University of Pittsburgh, Pittsburgh, PA 15232, USA. [6] University of Minnesota Genomics Center, University of Minnesota, Minneapolis, MN 55455, USA. [7] Diabetes Centre, Khoo Teck Puat Hospital, Singapore 768828, Singapore. [8] Saw Swee Hock School of Public Health, National University of Singapore, Singapore 117549, Singapore. [9] School of Public Health and Community Medicine, Hebrew University of Jerusalem, Jerusalem 12272, Israel. [10] Department of Medicine, Yong Loo Lin School of Medicine, National University of Singapore, Singapore 117597, Singapore. [11] Department of Epidemiology, Graduate School of Public Health, University of Pittsburgh, Pittsburgh, PA 15261, USA. [12] Health Systems and Services Research, Duke-NUS Medical School Singapore, Singapore 169857, Singapore. [13] Singapore Eye Research Institute, Singapore National Eye Centre, Singapore 169856, Singapore. [14] These authors contributed equally: Rajkumar Dorajoo, Xuling Chang, Resham Lal Gurung. [15] These authors jointly supervised this work: Jian-Min Yuan, Woon-Puay Koh, Chiea Chuen Khor, Chew-Kiat Heng. Correspondence and requests for materials should be addressed to C.C.K. (email: khorcc@gis.a-star.edu.sg) or to C.-K.H. (email: paehck@nus.edu.sg)

Telomeres are repetitive sequences of $(TTAGGG)_n$ that associate with shelterin proteins to cap the ends of the eukaryotic chromosomes. They protect individual chromosomes from degradation and inter-chromosomal fusion events[1]. Due to the inability of the DNA replication machinery to fully replicate the 5′ end of the lagging DNA strand, telomeres naturally shorten with each cell division. When critically short telomere length is reached (i.e., HayFlick limit), senescence and apoptosis mechanisms are induced in the cell[1]. Cellular telomere length is therefore a biological clock that determines the lifespan of a cell[2].

Most epidemiological studies have utilized telomere length measured in blood cells [i.e., leukocyte telomere length (LTL)] and within-individuals, this may be correlated with telomere length from multiple lineages and somatic cells from peripheral tissues[3]. Data have indicated that LTL shortens with age and are affected by gender and lifestyle factors[4,5]. Furthermore, shorter LTL is also associated with increased risks for numerous chronic diseases such as cardiovascular disease, respiratory disorders, type 2 diabetes mellitus (T2DM), liver diseases, metabolic syndrome, and neurodegenerative diseases, and overall mortality[4,6–10]. However the telomerase enzyme, which elongates telomeres and promotes cell survival and proliferation, is activated in most human cancers and longer LTL confers increased risks for several types of cancers[11,12]. These reports suggest a complex relationship between cellular telomere length, biological aging, and risks of various chronic diseases.

Heritability of LTL levels is approximately 30–60% and inter-individual LTL variation among adults are predominantly determined at birth[13–15]. Genome-wide association studies (GWAS) indicate that LTL is a complex polygenic trait. These genetic studies have identified at least eight different gene loci associated with LTL[16–22]. However, these have primarily been performed in populations of European ancestry and explained only a modest proportion of LTL variation (approximately 2% of phenotypic variation)[17]. Given that genetic determinants of telomere length may differ by ethnicity[2,23,24], it is likely that performing genetic studies in diverse populations could uncover additional genetic loci associated with LTL, as already seen in the South-Asian and African ancestry populations[22,25] and illuminate on cellular processes involved in human telomere length homeostasis.

Here, we undertake a GWAS of LTL in a relatively large Singaporean East-Asian (Southern Han Chinese) ethnic population (16,759 samples) and validate genome-wide significant associations in additional Singaporean Chinese samples (6337 samples). We further meta-analyze summary statistics from our current Singaporean Chinese datasets (23,096 samples in total) with data from large-scale European studies on LTL (37,505 samples). Our data expands on the genetic basis of human LTL. We further show that shorter LTL protects especially strongly against respiratory disease deaths in the Singapore Chinese population.

## Results

**Genome-wide LTL signals in the Singaporean Chinese.** We first performed a discovery GWAS analysis for associations with relative average telomere length in genomic DNA using 16,759 Southern Han Chinese samples from the Singapore Chinese Health Study (SCHS) and 6,407,959 SNPs (see Methods). We identified 7 genome-wide significant (score test $P < 5 \times 10^{-8}$) (Fig. 1) signals for LTL in this discovery GWAS analysis. The genomic inflation factor ($\lambda$) in the discovery GWAS was determined to be marginal ($\lambda = 1.043$); adjustment of association $P$-values based on the inflation factor did not

significantly affect the associations of the 7 identified genome-wide loci.

Three of these loci have not been previously reported and mapped to gene regions with known telomere maintenance function [lead SNPs rs3219104 in chromosome 1 (*PARP1*), rs28365964 in chromosome 8 (*TERF1*), and rs41293836 in chromosome 14 (*TINF2*)]. At 3 other loci, we identified lead SNP associations in the SCHS discovery GWAS that were in LD ($r^2 > 0.6$) with previously identified index SNPs from European GWAS studies (index SNPs rs10936599 in chromosome 3, rs7675998 in chromosome 4, and rs2736100 in chromosome 5, Supplementary Fig. 1)[16–21]. The 7th genome-wide significant signal was observed at the *OBFC1* gene locus in chromosome 10[17,18]. However, the lead SNP identified at this locus (rs12415148) was not in LD ($r^2 < 0.2$ in 1000G ASN panel) with the previously reported index SNP from European GWAS studies (rs9420907) (Fig. 2). Conditional probability analyses, adjusting our lead SNP (rs12415148) associations on the previous index SNP (rs9420907) genotypes (and vice versa), suggested that the signal identified in the SCHS study was independent from the previous signal at this *OBFC1* gene region (Supplementary Fig. 2).

**Replication of hits identified from discovery GWAS analysis.** Eleven loci surpassing LTL association $P < 1 \times 10^{-6}$ (score test) in the discovery stage analysis were brought forward for replication in a subset of samples from the SCHS (6337 additional Singaporean Chinese population samples, Table 1 and Supplementary Data 1). All 7 genome-wide significant signals identified from the discovery GWAS showed consistent and significant associations in the replication samples (score test $P_{Adj}$ between 0.045 and $2.45 \times 10^{-10}$, Table 1).

Two additional loci with suggestive levels of associations in the discovery stage (rs7776744 in chromosome 7 and rs227080 in chromosome 11) also showed significant evidence of replication (Table 1) and surpassed genome-wide significance on meta-analysis of discovery and replication stage samples (Table 1). The third suggestive signal in chromosome 20 was in the previously described *RTEL1* gene region (Table 1)[17,22]. However, our lead SNP (rs41309367) was not in LD with previously identified index SNPs (rs755017 and rs2297439) ($r^2 < 0.2$ in 1000G ASN panel) (Fig. 3). Conditional probability analyses using the Singaporean Chinese data indicated that significant associations at this locus were only lost after adjustments were made using genotypes of the newly identified lead SNP (rs41309367) and not the previously reported index SNPs (rs755017 and rs2297439) (Supplementary Fig. 3). One suggestive SNP (rs115346518 in chromosome 22) association however, showed inconsistent results in the replication samples and was likely to be a false-positive association (Table 1).

Therefore, within the SCHS Singaporean Chinese data we robustly confirmed previously reported index SNP associations with LTL at the *TERT*, *TERC*, and *NAF1* gene loci and further detected associations at 5 loci (*TINF2*, *PARP1*, *TERF1*, *ATM*, *POT1* gene loci) and 2 independent SNP associations at the known *OBFC1* and *RTEL1* gene loci (Table 1). We additionally looked up identified (Table 1) loci in the ENGAGE consortium's LTL GWAS performed using samples of European ancestry ($N = 37,505$) (https://downloads.lcbru.le.ac.uk/engage)[17]. At identified signals in the *PARP1* (rs3219104), *ATM* (rs227080), and *POT1* (rs7776744) gene loci, either the same lead SNP (rs3219104) or proxy SNPs (rs2267708 and rs645485) in LD ($r^2$ between 0.641 and 0.940 in 1000G CEU panel) with the lead SNPs identified in the Singaporean Chinese showed significant associations with LTL (meta-analysis

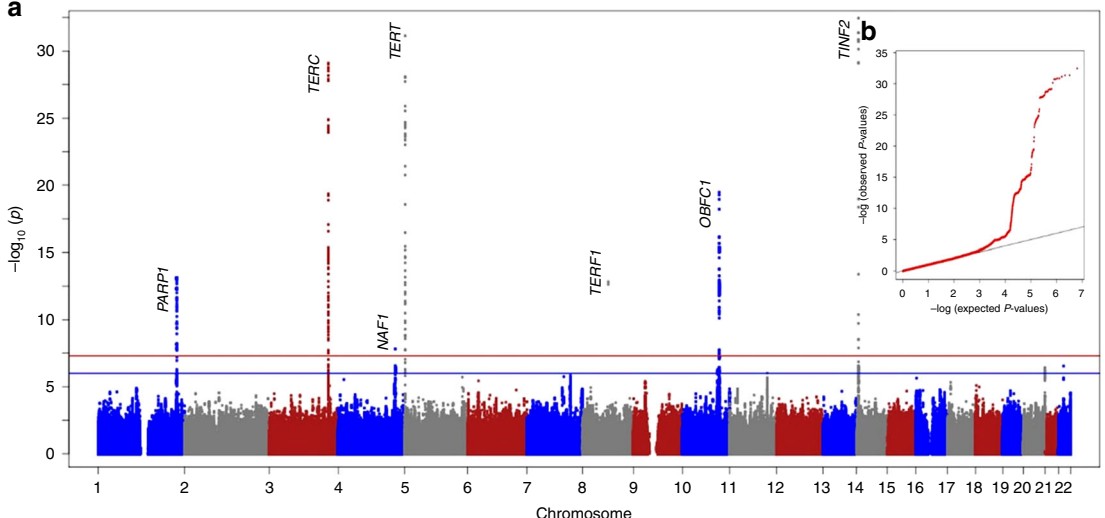

**Fig. 1** LTL associations in the SCHS discovery GWAS. **a** Seven hits at chromosome 1, 3, 4, 5, 8, 10, and 14 were identified beyond the genome-wide significance threshold (score test $P < 5 \times 10^{-8}$, red line) and 4 hits at chromosome 7, 11, 20, and 22 were identified beyond the suggestive significance threshold (score test $P < 1 \times 10^{-6}$, blue line). **b** QQ-plot of observed compared to expected $P$-values in the SCHS discovery GWAS indicated minimal inflation of study results ($\lambda = 1.043$)

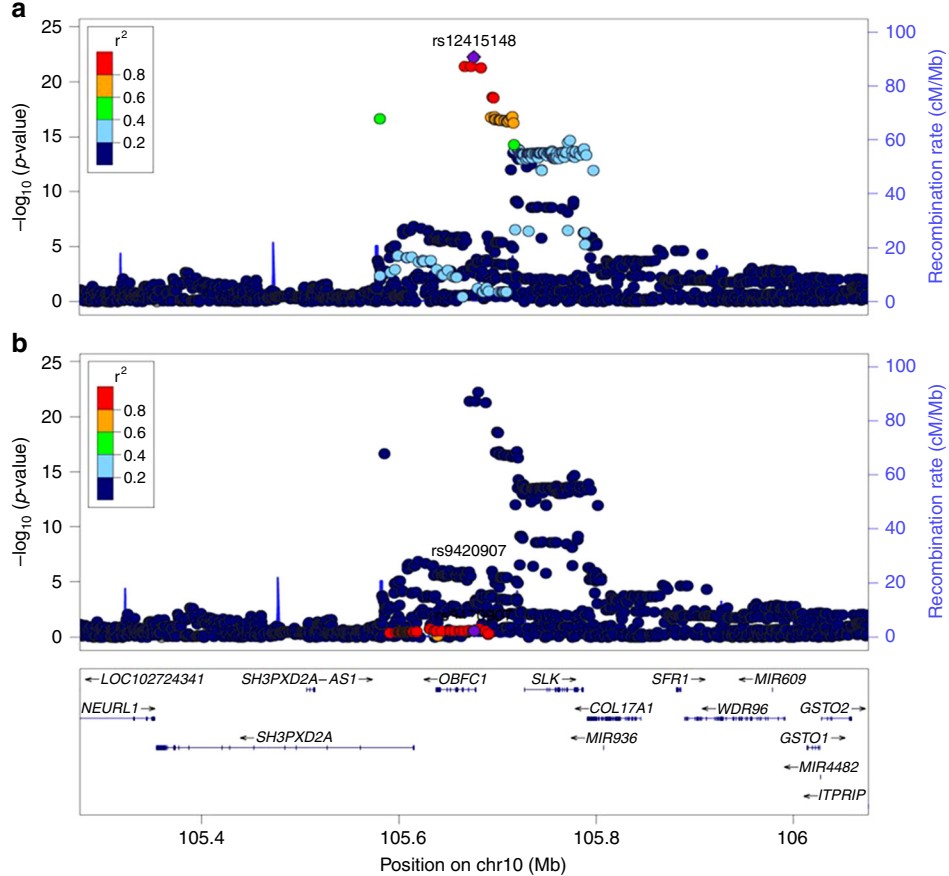

**Fig. 2** Regional SNP associations at the *OBFC1* gene locus in the SCHS discovery GWAS. **a** Association of lead SNP identified in the SCHS (rs12415148). **b** Association of previously identified index SNP from European GWAS studies (rs9420907). Lead SNP indicated as purple diamonds. LD ($r^2$) data of SNPs based on ASN panels of 1000Genome database. Plots plotted using LocusZoom (http://csg.sph.umich.edu/locuszoom/)

$P$ between $5.31 \times 10^{-5}$ and $9.30 \times 10^{-6}$, Table 1 and Supplementary Fig. 4). The 2 other loci identified in the Singaporean Chinese GWAS were monomorphic (rs28365964) or rare (MAF < 1%, rs41293836) in the European populations and no proxy SNPs in

strong LD in Europeans were available for evaluation (best proxy SNP at the *TINF2* locus in chromosome 14 was rs3742506, however LD between rs3742506 and the rs41293836 lead SNP was weak, $r^2 = 0.35$ in 1000G CEU panel).

**Table 1 Summary statistics of genome-wide hits identified within the SCHS study**

| SNP | Chr | Position | Genes | Comment | Discovery (N = 16,759) | | | Replication (N = 6337) | | Meta (N = 23,096) | | ENGAGE (N = 37,505) | | | | Meta (N = 60,061) | |
|---|---|---|---|---|---|---|---|---|---|---|---|---|---|---|---|---|---|
| | | | | | TA | Beta | P | Beta | $P_{Adj}$ | Beta | P | Proxy SNP | TA | Beta | P | Beta | P |
| rs41293836 | 14 | 24721327 | TINF2 | Novel locus | C | −0.242 | $3.53 \times 10^{-33}$ | −0.21 | $5.34 \times 10^{-10}$ | −0.233 | $2.47 \times 10^{-42}$ | NA | NA | NA | NA | NA | NA |
| rs7705526 | 5 | 1285974 | TERT | Known locus | C | −0.129 | $7.35 \times 10^{-32}$ | −0.091 | $1.36 \times 10^{-6}$ | −0.118 | $2.61 \times 10^{-38}$ | rs10936599 | T | −0.097 | $2.23 \times 10^{-31}$ | −0.107 | $3.53 \times 10^{-67}$ |
| rs2293607 | 3 | 169482335 | TERC | Known locus | C | −0.122 | $2.99 \times 10^{-29}$ | −0.114 | $2.51 \times 10^{-10}$ | −0.12 | $7.57 \times 10^{-39}$ | rs2736100 | C | −0.078 | $4.05 \times 10^{-19}$ | −0.093 | $5.35 \times 10^{-48}$ |
| rs12415148 | 10 | 105680586 | OBFC1 | Known locus, independent SNP | T | −0.218 | $3.34 \times 10^{-20}$ | −0.166 | $9.38 \times 10^{-5}$ | −0.204 | $2.78 \times 10^{-25}$ | NA | NA | NA | NA | NA | NA |
| rs3219104 | 1 | 226562621 | PARP1 | Novel locus | A | −0.08 | $7.70 \times 10^{-14}$ | −0.06 | $4.61 \times 10^{-3}$ | −0.074 | $2.23 \times 10^{-16}$ | NA | A | −0.039 | $5.31 \times 10^{-5}$ | −0.057 | $2.38 \times 10^{-18}$ |
| rs28365964 | 8 | 73920883 | TERF1 | Novel locus | T | −0.305 | $1.47 \times 10^{-13}$ | −0.186 | 0.039 | −0.27 | $6.96 \times 10^{-15}$ | NA | NA | NA | NA | NA | NA |
| rs10857352 | 4 | 164101482 | NAF1 | Known locus | A | −0.068 | $1.52 \times 10^{-8}$ | −0.057 | 0.045 | −0.064 | $4.85 \times 10^{-9}$ | NA | A | −0.036 | $1.30 \times 10^{-5}$ | −0.045 | $3.63 \times 10^{-12}$ |
| rs41309367 | 20 | 62309554 | RTEL1 | Known locus, independent SNP | T | −0.061 | $4.14 \times 10^{-7}$ | −0.053 | 0.081 | −0.058 | $1.16 \times 10^{-8}$ | NA | NA | NA | NA | NA | NA |
| rs227080 | 11 | 108247888 | ATM | Novel locus | G | −0.054 | $9.15 \times 10^{-7}$ | −0.074 | $2.95 \times 10^{-4}$ | −0.06 | $1.87 \times 10^{-10}$ | rs645485 | G | −0.032 | $9.30 \times 10^{-6}$ | −0.039 | $6.55 \times 10^{-12}$ |
| rs7776744 | 7 | 124599749 | POT1 | Novel locus | G | −0.051 | $8.38 \times 10^{-7}$ | −0.073 | $2.24 \times 10^{-4}$ | −0.058 | $2.51 \times 10^{-10}$ | rs2267708 | C | −0.033 | $9.73 \times 10^{-6}$ | −0.039 | $2.09 \times 10^{-11}$ |
| rs115346518 | 22 | 31113059 | OSBP2 | Non-significant locus | G | −0.091 | $2.87 \times 10^{-7}$ | 0.008 | 1 | −0.063 | $2.91 \times 10^{-5}$ | NA | NA | NA | NA | NA | NA |

TA: test allele; SCHS replication $P_{Adj}$: score test P adjusted for 11 tests; discovery score test P adjusted for genomic inflation factor ($\lambda = 1.043$). Proxy SNP: SNPs from ENGAGE dataset that were in LD ($r^2 > 0.6$ in CEU 1000G panel) with lead SNPs identified in the SCHS

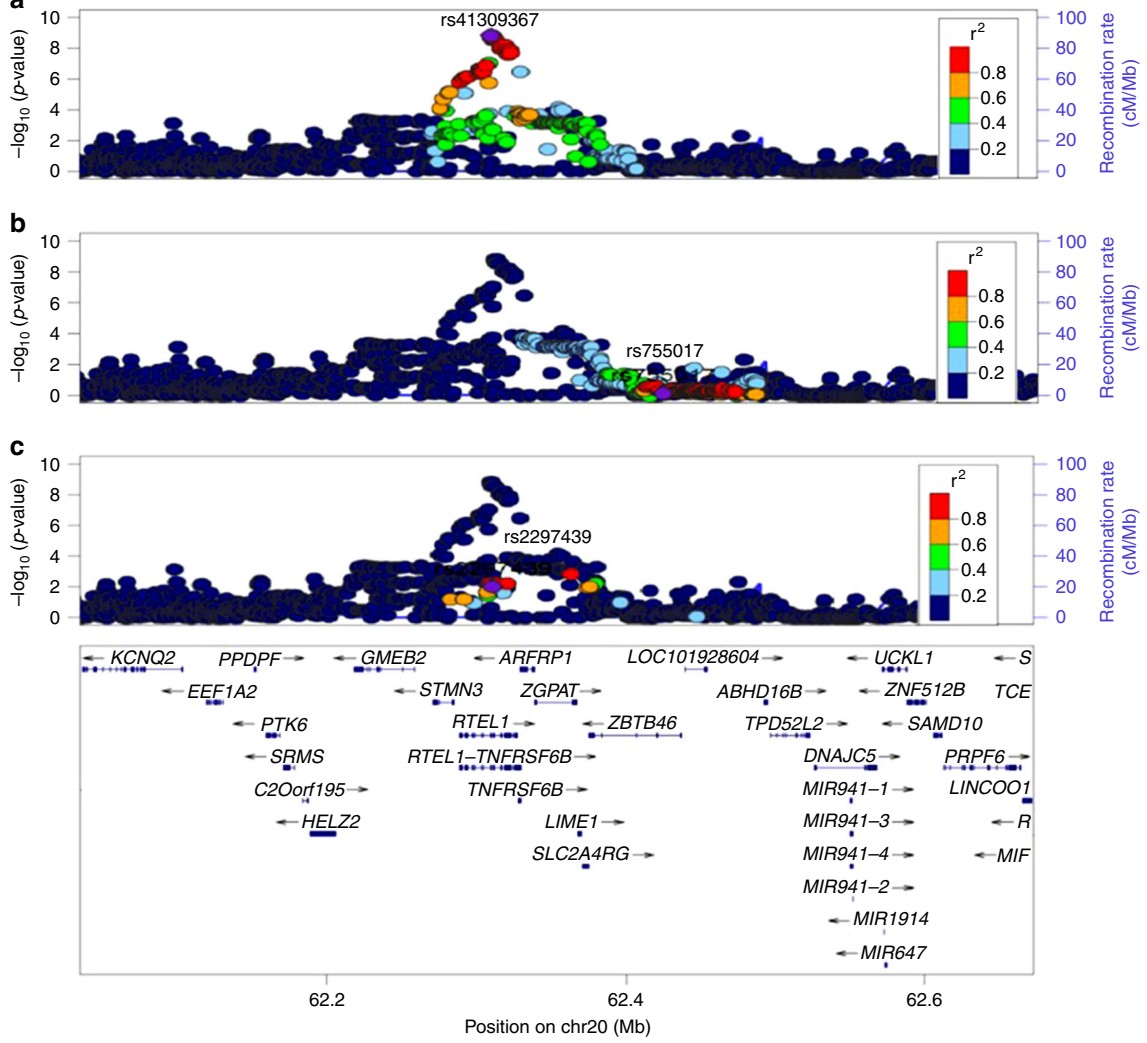

**Fig. 3** Regional SNP associations at the *RTEL1* gene locus in the SCHS discovery GWAS. **a** Association of lead SNP identified in the SCHS (rs41309367). **b** Association of previously identified index SNP from European GWAS studies (rs755017). **c** Association of previously identified index SNP from South Asian GWAS studies (rs2297439). Lead SNP indicated as purple diamonds. LD ($r^2$) data of SNPs based on ASN panels of 1000Genome database. Plots plotted using LocusZoom (http://csg.sph.umich.edu/locuszoom/)

**Table 2 Summary statistics of genome-wide hits identified after trans-ethnic meta-analysis**

| SNP | Chr | Position | Genes | Comment | Discovery (N = 16,759) | | | Replication (N = 6337) | | Meta (N = 23,096) | | ENGAGE (N = 37,505) | | | Meta (N = 60,061) | |
|---|---|---|---|---|---|---|---|---|---|---|---|---|---|---|---|---|
| | | | | | TA | Beta | P | Beta | P | Beta | P | TA | Beta | P | Beta | P |
| rs7095953 | 10 | 101274425 | NKX2-3 | Novel locus | C | −0.052 | $1.12 \times 10^{-6}$ | −0.034 | 0.046 | −0.047 | $2.16 \times 10^{-7}$ | C | −0.033 | $8.83 \times 10^{-3}$ | −0.042 | $9.59 \times 10^{-9}$ |
| rs7253490 | 19 | 22293706 | ZNF257 | Known locus | C | −0.046 | $4.41 \times 10^{-5}$ | −0.011 | 0.525 | −0.036 | $1.44 \times 10^{-4}$ | C | −0.047 | $1.66 \times 10^{-9}$ | −0.043 | $1.38 \times 10^{-12}$ |
| rs2967374 | 16 | 82209861 | MPHOSPH6 | Novel locus | G | −0.061 | $3.36 \times 10^{-5}$ | −0.043 | 0.067 | −0.056 | $7.16 \times 10^{-6}$ | G | −0.045 | $2.63 \times 10^{-6}$ | −0.049 | $1.00 \times 10^{-11}$ |
| rs2302588 | 14 | 73404752 | DCAF4 | Known locus, independent SNP | G | −0.049 | $2.11 \times 10^{-4}$ | −0.027 | 0.188 | −0.042 | $1.27 \times 10^{-4}$ | G | −0.071 | $6.15 \times 10^{-7}$ | −0.053 | $8.57 \times 10^{-9}$ |
| rs1001761 | 18 | 662103 | TYMS | Novel locus | A | −0.037 | $1.68 \times 10^{-3}$ | −0.055 | $3.66 \times 10^{-3}$ | −0.042 | $2.70 \times 10^{-5}$ | A | −0.029 | $6.22 \times 10^{-5}$ | −0.034 | $1.06 \times 10^{-8}$ |
| rs11890390 | 2 | 54485682 | ACYP2 | Known locus | C | −0.038 | $6.61 \times 10^{-3}$ | −0.047 | 0.033 | −0.040 | $6.10 \times 10^{-4}$ | C | −0.054 | $1.63 \times 10^{-7}$ | −0.048 | $5.18 \times 10^{-10}$ |

TA: test allele; discovery score test P adjusted for genomic inflation factor ($\lambda = 1.043$). Proxy SNP: SNPs from ENGAGE dataset that were in LD ($r^2 > 0.6$ in CEU 1000G panel) with lead SNPs identified in the SCHS

**Trans-ethnic meta-analyses**. We next conducted a meta-analysis of all Singaporean Chinese (N = 23,096) and European ancestry LTL data (N = 37,505) (https://downloads.lcbru.le.ac.uk/engage)[17]. This trans-ethnic meta-analysis identified 6 additional loci surpassing genome-wide significance (Table 2, meta-data of all genome-wide SNPs presented in Supplementary Data 2).

Three out of the 6 loci which emerged after trans-ethnic meta-analysis were not previously reported (Table 2). Two other loci were previously identified in European GWAS studies for LTL[17]. At the ZNF208, ZNF257, ZNF676 gene locus in chromosome 19, the identified lead SNP (rs7253490) was in LD with the previously reported index SNP at this locus (rs8105767, $r^2 = 0.785$ in 1000G CHB panel, Supplementary Fig. 5) and at the ACYP2 gene locus in chromosome 2, the identified lead SNP (rs11890390) was in LD with the previously reported index SNP at this locus (rs11125529, $r^2 = 0.862$ in 1000G CHB panel, Supplementary Fig. 5). The last signal detected was at the previously identified DCAF4 gene locus in chromosome 14. However the lead SNP identified (rs2302588) was independent from the previously identified index SNP (rs2535913, $r^2 < 0.062$ in 1000G panels) (Fig. 4 and Supplementary Fig. 6).

Increased age and males were highly associated with reduced LTL in our dataset, explaining approximately 6.30% and 1.29% of phenotypic variation (Supplementary Table 1). In total the 16 genome-wide loci identified explained approximately 3.98% of the phenotypic variation in LTL among the samples from the combined SCHS study (Supplementary Table 2), roughly doubling the variance explained by previous studies[17].

**Additional look-up in diabetic samples**. The 16 genome-wide SNPs from the study were further evaluated in 1602 Singaporean Chinese type 2 diabetic subjects. Of these, 12 were directionally consistent in the type 2 diabetic cases (binomial P = 0.028) and the top hit at chromosome 14 (rs41293836) showed statistically significant association with LTL (Supplementary Table 3).

**Functional annotations and gene set enrichments**. FUMA GWAS (Functional Mapping and Annotation of Genome Wide Association Studies)[26] and ingenuity pathway analyses were utilized to functionally annotate all significant SNPs identified in the study and evaluate mapped genes in a biological context. A total of 2020 candidate SNPs in LD ($r^2 > 0.6$ and MAF > 0.1% in 1000G ASN panel) with the 16 genome-wide SNPs identified in the study and 268 regional genes (within 200 kb from lead SNPs) were mapped (Supplementary Fig. 7). Identified candidate SNPs were predominantly located in intronic and intergenic regions (Supplementary Fig. 7). However, strong Combined Annotation Dependent Depletion (CADD) annotations[27] (CADD > 20, top

1% of deleterious variants in the human genome) and potentially protein damaging predictions were identified at 2 SNPs that were in LD with GWAS lead SNPs ($r^2 > 0.99$, Supplementary Data 3). These include the exonic SNP at the DCAF4 gene locus (rs3815460, 345S>C) that was near a WD40 domain (UniProtKB) and another exonic SNP (rs1136410, 762V>A) at the PARP1 gene locus (CADD = 32, top 0.1% of deleterious variants in the human genome) that has been reported to affect the catalytic domain of PARP1 through the binding of $NAD^+$ at residue 762[28].

For the 268 regional LTL genes, 454 significant quantitative trait loci [expression QTL (eQTL), methylation QTL (mQTL), and histone modification QTL (hQTL), linear regression Bonferroni corrected $P < 5 \times 10^{-8}$] SNPs were identified from the BLUEPRINT (BLUEPRINT of Haematopoietic Epigenomes) blood immune cell-type specific (monocytes, T cells, and neutrophils) epigenome data[29] (Supplementary Data 4). Four LTL GWAS lead SNPs from the study were in LD with four QTL SNPs ($r^2$ between 0.985 and 0.916, 1000G ASN panel). These include hQTL SNPs, rs2911429 and rs9969187, affecting histone peaks at MPHOSPH6 in neutrophils and POT1 in monocytes, respectively. The rs7253490 GWAS lead SNP was in LD with an eQTL SNP (rs17554725, $r^2 = 0.944$, 1000G ASN panel) that affects ZNF257 gene expression in both monocytes and T cells. The A allele of rs17554725 was observed to decrease LTL levels in our data (beta = −0.0396, meta-analysis $P = 5.24 \times 10^{-5}$, Supplementary Data 1) and decreased ZNF257 expression in both monocytes and T cells (Supplementary Data 4). The rs227080 GWAS lead SNP was in LD with another eQTL SNP (rs660429, $r^2 = 0.916$, 1000G ASN panel) that affects ATM expression in T cells. Rs660429 was also a significant eQTL in T cells for ATM expression in the DICE (database of immune cell expression, expression quantitative trait loci and epigenomics) study[30] (Supplementary Data 5). The C allele of rs660429 was observed to decrease LTL levels (beta = −0.0526, meta-analysis $P = 9.03 \times 10^{-9}$) in our data and increase ATM expression in T cells (Supplementary Data 1 and 4). Co-localization analyses at these regions indicated moderate to high probability for shared causal variant between these QTL association signals and LTL GWAS association signals [posterior probability between 61.71% and 96.32% (Supplementary Data 4)].

Gene Ontology (GO) term analyses and canonical pathway analyses based on the 268 mapped regional genes indicated significant enrichments in pathways associated with telomere and DNA repair such as (1) telomere maintenance via telomere lengthening and telomerase activity, (2) base excision repair, and (3) DNA double strand break repair (Supplementary Figs. 8–10). GO cellular component analyses also indicated that specific molecules identified from the study were implicated at different subcellular regions (for e.g., OBFC1, TINF2, TERT, POT1, and

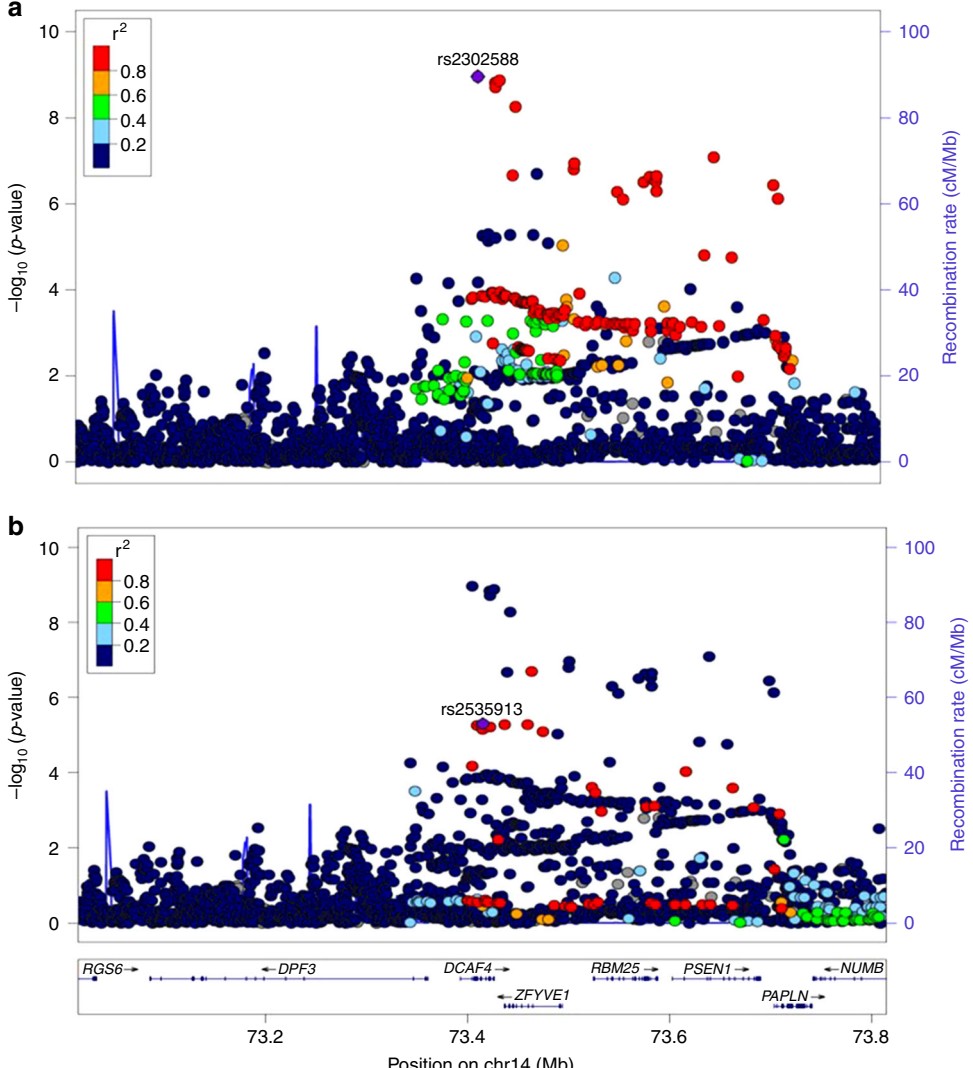

**Fig. 4** Regional SNP associations at the *DCAF4* gene locus in the trans-ethnic meta-analysis. **a** Association of lead SNP identified in the study (rs2302588). **b** Association of previously identified index SNP from European GWAS studies (rs2535913). Lead SNP indicated as purple diamonds. LD ($r^2$) data of SNPs based on ASN panels of 1000Genome database. Plots plotted using LocusZoom (http://csg.sph.umich.edu/locuszoom/)

*TERF1* were components of the telomere cap complex, while *PARP1*, *ATM*, *TOX4*, *THOC1*, and *SMCHD1* were associated with chromosome telomeric regions, Supplementary Fig. 9A). The LTL regional genes were also significantly enriched in previous GWAS loci associated with telomere length as well as in several other diseases and traits (Fig. 5). Repeating this analysis after exclusion of strong telomere-related genes (Tables 1 and 2), expectedly abolished associations with loci implicated in telomere length genetic studies. Moreover, enrichments in gene loci reported for melanoma, glioma, non-glioblastoma glioma, glioblastoma, thyroid cancer, lung cancer, uterine fibrosis, interstitial lung disease, breast cancer in *BRACA1* mutation carriers, response to serotonin reuptake inhibitors and depression and BMI change over time were lost, perhaps indicating that telomere-related genes were also relevant in these diseases and traits (Supplementary Fig. 11). Additionally, we overlapped the 268 regional LTL genes with those previously reported for gene expression and methylation clocks of aging[31,32]. One gene (*UCKL1*) overlapped with methylation-age-associated gene loci [Supplementary Data 6]. Nineteen LTL regional genes overlapped with expression-age-associated gene loci[31] and indicated a

significant enrichment in the DNA double-strand break repair by non-homologous end joining pathway (involving *ATM* and *PARP1*, Fisher's exact test $P_{Adj} = 0.013$) (Supplementary Data 6 and Supplementary Fig. 12).

**Association of loci with mortality due to chronic diseases.** Longer LTL in the SCHS dataset was associated with reduced risk of death [all-cause mortality HR = 0.953 (0.928–0.979), Cox regression $P_{Adj} = 2.60 \times 10^{-3}$], cardiovascular diseases deaths [HR = 0.934 (0.883–0.982), Cox regression $P_{Adj} = 0.048$] and especially mortality due to respiratory diseases [HR = 0.854 (0.804–0.906), Cox regression $P_{Adj} = 1.13 \times 10^{-6}$] (Table 3). Stratifying respiratory disease deaths into those due to infections (pneumoniae and influenza) and chronic obstructive pulmonary disease (COPD) indicated that longer LTL was equally protective for both these outcomes (Table 3). Longer LTL was also nominally associated with increased risk of mortality due to cancer, although this association was not significant after corrections for multiple tests (Table 3).

We evaluated if the 16 genetic signals, both individually and through a weighted gene risk score (wGRS) for reduced LTL, were

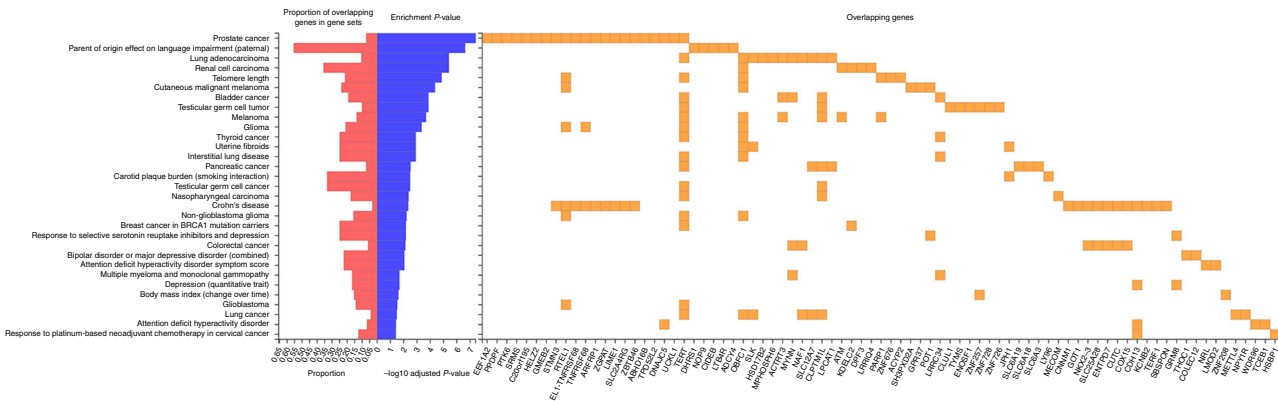

**Fig. 5** Gene set enrichment of mapped regional genes for LTL association in previous GWAS studies. Enrichment *P*-values adjusted for Bonferroni corrections and proportion indicates the percentage of input LTL associated genes that overlap with genes implicated in previous GWAS traits and diseases

### Table 3 Association of *Z*-score for LTL with various mortalities in the SCHS dataset

| | Cases/Controls | HR (95% CI) | *P* | *P*<sub>Adj</sub> |
|---|---|---|---|---|
| Cancer mortalities | 2190/20,942 | 1.047 (1.002–1.095) | 0.039 | 0.234 |
| Cardiovascular disease mortalities | 1670/21,462 | 0.934 (0.883–0.982) | $7.93 \times 10^{-3}$ | 0.048 |
| All respiratory disease mortalities | 1223/21,909 | 0.854 (0.804–0.906) | $1.88 \times 10^{-7}$ | $1.13 \times 10^{-6}$ |
| Pneumoniae/influenza mortalities | 1027/22,105 | 0.867 (0.812–0.925) | $1.48 \times 10^{-5}$ | $8.88 \times 10^{-5}$ |
| Chronic obstructive pulmonary disease mortalities | 196/22,936 | 0.781 (0.671–0.908) | $1.34 \times 10^{-3}$ | $8.04 \times 10^{-3}$ |
| All mortalities | 6035/17,097 | 0.953 (0.928–0.979) | $4.33 \times 10^{-4}$ | $2.60 \times 10^{-3}$ |

Cox regression *P* adjusted for age, sex, PC1–3, BMI, and smoking status (never smokers vs current/ex-smokers). *P*<sub>Adj</sub>: association *P* adjusted for 6 tests

associated with mortality in the SCHS dataset. The wGRS for reduced LTL was not associated with mortalities (Table 4). However, we detected a significant association between the LTL reducing C allele of rs7253490 with increased risk of mortality due to respiratory disease [HR = 1.181 (1.084–1.285), Cox regression $P_{Adj} = 2.71 \times 10^{-3}$] (Table 4). Stratifying respiratory deaths indicated that this SNP was significantly associated with deaths due to respiratory infections [HR = 1.173 (1.069–1.287), Cox regression $P = 7.44 \times 10^{-4}$, Supplementary Table 4]. Mediation analysis was performed to evaluate if the respiratory infection mortality effect of rs7253490 was mediated through LTL. Although the effect of rs7253490 on respiratory infection deaths mediated through LTL was significant (linear regression $P = 0.014$), the proportion mediated was modest (3.28%) (Supplementary Table 5 and Supplementary Fig. 13).

### Discussion

In this genetic study for LTL association, through an analysis within the Singaporean Chinese population samples ($N = 23,096$) and through larger-scale trans-ethnic meta-analysis ($N = 60,061$), we identified 16 genome-wide loci, including 3 independent SNP associations at previously identified loci (*OBFC1*, *RTEL1*, and *DCAF4* gene loci). Five known loci (*TERC* locus in chromosome 3, *TERT* locus in chromosome 5, *NAF1* locus in chromosome 4, *ZNF208*, *ZNF257*, *and ZNF676* locus in chromosome 19, and *ACYP2* locus in chromosome 2) were also detected at genome-wide levels of significance in the study. Our data therefore not only corroborates previous genetic findings for LTL in the Chinese ethnic population, but also doubles the number of loci associated with LTL and provides additional insights into the genetic control of human telomere homeostasis mechanisms.

Our results firstly have demonstrated the value of utilizing non-European ethnic populations for genetic association studies.

Loci that were monomorphic or rare in the European populations but polymorphic and common in the Chinese population could have been missed even in large-scale studies that analyzed only European ancestry samples (i.e., rs28365964 at the *TERF1* locus in chromosome 8 and rs41293836 at the *TINF2*/*TGM1* loci in chromosome 14). Loci where independent SNP associations (from previous index SNPs from European GWAS studies) were identified also highlighted the potential ethnic differences in LD structure or that multiple independent causal events at the same locus might have occurred. Differences in risk allele frequencies between the Chinese and European ethnic populations and/or the increased samples sizes may have also improved power in the study to identify trans-ethnic loci (for e.g., rs3219104 at the *PARP1* locus in chromosome 1, rs227080 at the *ATM* locus in chromosome 11, rs2967374 at the *MPHOSPH6* locus in chromosome 16, rs7776744 at the *POT1* locus in chromosome 7 and rs1001761 at the *TYMS* locus in chromosome 18, rs7095953 at the *NKX2-3* locus in chromosome 10, and rs2302588 at the *DCAF4* locus in chromosome 14). It should be noted that a number of these trans-ethnic loci that were transferable between the Chinese and European populations were first reported at suggestively significant association levels (meta-analysis *P* between $8.83 \times 10^{-3}$ and $1.63 \times 10^{-7}$) in earlier GWAS for LTL performed using European samples only[17].

At the *PARP1* and *DCAF4* loci, potentially deleterious functional exonic variants were identified. At two other LTL GWAS loci, identified lead SNPs were in strong LD with circulating immune cell-type eQTL SNPs, potentially implicating *ATM* and *ZNF257* as likely functional genes at these GWAS loci. Two other identified histone modification peaks also overlapped with the *MPHOSPH6* and *POT1* gene regions. Whether or not the two histone peaks directly result in regulation of *MPHOSPH6* and *POT1* would require further biological investigations.

**Table 4 Association of LTL reducing SNPs and weighted gene-risk score with mortalities in the SCHS**

| SNP | TA | Genes | Cancer mortalities (2190 cases/20,942 controls) | | | Cardiovascular mortalities (1670 cases/21,462 controls) | | | Respiratory mortalities (1223 cases/21,909 controls) | | | All mortalities (6035 cases/17,097 controls) | | |
|---|---|---|---|---|---|---|---|---|---|---|---|---|---|---|
| | | | HR (95% CI) | $P$ | $P_{Adj}$ | HR (95% CI) | $P$ | $P_{Adj}$ | HR (95% CI) | $P$ | $P_{Adj}$ | HR (95% CI) | $P$ | $P_{Adj}$ |
| wGRS | | | 0.979 (0.961-0.997) | 0.027 | 0.558 | 1.012 (0.991-1.034) | 0.247 | 1 | 1.013 (0.988-1.038) | 0.298 | 1 | 1.001 (0.989-1.011) | 0.929 | 1 |
| rs1001761 | A | TYMS | 0.968 (0.906-1.035) | 0.351 | 1 | 0.977 (0.905-1.055) | 0.557 | 1 | 1.038 (0.948-1.137) | 0.414 | 1 | 0.980 (0.941-1.020) | 0.330 | 1 |
| rs10857352 | A | NAF1 | 1.009 (0.943-1.079) | 0.787 | 1 | 1.006 (0.931-1.087) | 0.870 | 1 | 0.956 (0.873-1.048) | 0.343 | 1 | 0.998 (0.958-1.039) | 0.930 | 1 |
| rs11890390 | C | ACYP2 | 1.028 (0.949-1.114) | 0.493 | 1 | 1.051 (0.958-1.1538) | 0.290 | 1 | 0.944 (0.850-1.048) | 0.286 | 1 | 0.998 (0.952-1.048) | 0.967 | 1 |
| rs12415148 | T | OBFC1 | 1.141 (0.986-1.320) | 0.075 | 1 | 0.884 (0.762-1.026) | 0.107 | 1 | 0.814 (0.688-0.964) | 0.017 | 0.359 | 0.989 (0.911-1.074) | 0.806 | 1 |
| rs227080 | G | ATM | 0.951 (0.893-1.013) | 0.123 | 1 | 0.952 (0.886-1.022) | 0.179 | 1 | 1.040 (0.957-1.130) | 0.353 | 1 | 0.976 (0.940-1.014) | 0.222 | 1 |
| rs2293607 | C | TERC | 0.969 (0.913-1.029) | 0.313 | 1 | 1.069 (0.998-1.145) | 0.055 | 1 | 0.993 (0.916-1.076) | 0.869 | 1 | 1.021 (0.985-1.059) | 0.239 | 1 |
| rs2302588 | G | DCAF4 | 0.914 (0.851-0.982) | 0.015 | 0.317 | 0.949 (0.874-1.032) | 0.226 | 1 | 1.004 (0.909-1.107) | 0.936 | 1 | 0.954 (0.913-0.996) | 0.036 | 0.755 |
| rs28365964 | T | TERF1 | 0.814 (0.654-1.012) | 0.064 | 1 | 1.049 (0.793-1.388) | 0.736 | 1 | 1.001 (0.726-1.379) | 0.995 | 1 | 0.908 (0.791-1.043) | 0.173 | 1 |
| rs2967374 | G | MPHOSPH6 | 0.989 (0.911-1.074) | 0.808 | 1 | 0.988 (0.900-1.086) | 0.816 | 1 | 1.007 (0.901-1.125) | 0.897 | 1 | 0.969 (0.922-1.018) | 0.215 | 1 |
| rs3219104 | A | PARP1 | 0.968 (0.912-1.027) | 0.284 | 1 | 1.072 (1.002-1.147) | 0.043 | 1 | 1.041 (0.962-1.127) | 0.308 | 1 | 1.010 (0.975-1.047) | 0.567 | 1 |
| rs41293836 | C | TINF2 | 1.033 (0.922-1.157) | 0.572 | 1 | 1.038 (0.911-1.181) | 0.572 | 1 | 1.083 (0.930-1.263) | 0.302 | 1 | 1.065 (0.994-1.141) | 0.073 | 1 |
| rs41309367 | T | RTEL1 | 1.009 (0.944-1.079) | 0.776 | 1 | 0.997 (0.923-1.077) | 0.941 | 1 | 1.086 (0.994-1.186) | 0.066 | 1 | 1.018 (0.977-1.060) | 0.379 | 1 |
| rs7095953 | C | NKX2-3 | 0.980 (0.922-1.041) | 0.515 | 1 | 1.028 (0.959-1.101) | 0.429 | 1 | 1.006 (0.928-1.091) | 0.874 | 1 | 0.998 (0.962-1.035) | 0.937 | 1 |
| rs7253490 | C | ZNF257 | 1.013 (0.951-1.078) | 0.681 | 1 | 1.001 (0.931-1.073) | 0.996 | 1 | 1.181 (1.084-1.285) | $1.29 \times 10^{-4}$ | $2.71 \times 10^{-3}$ | 1.044 (1.006-1.085) | 0.023 | 0.480 |
| rs7705526 | C | TERT | 0.907 (0.830-0.991) | 0.031 | 0.644 | 1.047 (0.948-1.156) | 0.362 | 1 | 1.066 (0.950-1.197) | 0.275 | 1 | 0.995 (0.944-1.048) | 0.85 | 1 |
| rs7776744 | G | POT1 | 0.988 (0.931-1.049) | 0.708 | 1 | 1.037 (0.968-1.111) | 0.295 | 1 | 0.957 (0.883-1.037) | 0.290 | 1 | 1.002 (0.966-1.039) | 0.89 | 1 |

Test allele: LTL reducing allele; $P_{Adj}$: Cox regression $P$ adjusted for 21 tests. Associations adjusted for age, sex, PC1-3, BMI, and smoking status (never-smokers vs current/ex-smokers)

Similar to previously reported genes identified for LTL (i.e., TERT, TERC, and NAF1)[17], almost all loci identified in our study lie in or near genes that perform prominent roles in preserving telomere structure, regulating telomere length, and in DNA repair pathways. TINF2, POT1, and TERF1 are components of the telomere shelterin protein complex and regulate telomerase activity[33–38]. Correspondingly, these genes were identified to be located at the telomere cap complex in GO cellular component enrichment analyses. Severe deleterious mutations in TINF2 have also been identified in patients with dyskeratosis congenita that is characterized by very short telomeres[33,34]. POT1, together with another shelterin protein complex protein, TPP1, forms a complex with telomeric DNA that increases the activity and processivity functions of the telomerase enzyme[37,38]. TERF1 is a negative regulator of telomere length and post-translational modifications (e.g., ADP-ribosylation) at TERF1 affects its DNA binding capacity[35,36]. One such protein that affects TERF1 activity at telomeres is PARP1[39–41]. PARP1 has been described to ribosylate TERF1, affecting its ability to bind to telomeric DNA and therefore, acts as a positive regulator for telomere elongation.

Chromosome ends are protected by shelterin. One of the known key functions of shelterin is in preventing the activation of three DNA damage response enzymes ATM, ATR kinases, and PARP1[42,43]. It is noteworthy that variants in genes encoding for two of the three DNA damage response enzymes have been identified in this study to be associated with LTL (i.e., ATM and PARP1). Overlap of ATM and PARP1 genes with gene loci previously identified as components of gene expression aging clocks[31], recapitulates the importance of DNA repair mechanisms (for e.g., DNA double-strand break repair) in LTL homeostasis as well as in the overall human aging process. The TYMS gene also catalyzes the methylation of deoxyuridylate to deoxythymidylate and maintains the dTMP pool to enable efficient DNA replication and repair processes[44]. Other regional genes identified in the study such as TOX4, THOC1, and SMCHD1 were also involved in

DNA damage surveillance and repair mechanisms and were observed to localize at the telomeric region[45–47].

The role of NKX2-3 and MPHOSPH6 in telomere length maintenance is however not directly obvious. NKX2-3 plays a prominent role in immune and inflammatory responses and has been previously been implicated in inflammatory bowel disease pathogensis[48]. NKX2-3 may also be involved as a TERT-response gene in cellular reprogramming[49]. MPHOSPH6 is involved in the association of exosomes to the pre-rRNA[50] and has been shown to affect cell cycle and ovary development in shrimps[51].

It has also been reported that shorter LTL is associated with diabetes status, including the Chinese ethnicity[52]. Comparing relative mean LTL in the SCHS population samples and the diabetic cases (SMART2D and DN datasets) indicated that LTL was generally lower in the diabetic patients as compared to the SCHS population samples (Supplementary Table 6). Nevertheless, at these specific loci detected for LTL associations, direction of effects were generally consistent in the diabetic samples evaluated and the top hit (rs41293836 at chromosome 14) identified in the SCHS population samples showed significant association in the diabetic samples as well.

Previous Mendelian randomization studies have reported a causal effect between longer LTL and increased risks of cancer and decreased risks of non-cancer disorders[7]. In this study, we report similar trends between measured LTL and deaths due to cancer, respiratory, and cardiovascular diseases in the Singaporean Chinese population. The association between shorter LTL and risks for respiratory disease deaths was particularly strong in our dataset. While the associations between shortened LTL and increased COPD risk has been previously reported[6,7] and confirmed in our data, we additionally observe that shorter LTL and the rs7253490 SNP was also associated with deaths due to respiratory infections. Our eQTL look-up indicated that a LTL reducing SNP in LD with rs7253490 decreased ZNF257 gene expression in T cells and monocytes, implicating ZNF257 as a

potential functional gene at this GWAS locus and ZNF257 may function as a transcription factor that affects downstream genes linked to immune response[53]. These data suggest that a sub-proportion of the genetic control of human telomere length homeostasis may be pathological and perhaps indicates that LTL and rs7253490 may be markers of immune competence[54]. At the same time however, it is important to note that much of the respiratory infection mortality effects for rs7253490 was not strongly mediated through LTL, signifying potential pleotropic effects at this genetic locus. Interaction analyses with multiple lifestyle factors (i.e., smoking status, alcohol consumption, physical activity, and obesity levels) however, did not identify interactions that modified the associations between rs7253490 and respiratory infection deaths (Supplementary Table 7) in our study samples.

One limitation with the GWAS for LTL associations within the SCHS population samples was the smaller number of subjects in the replication stage (as compared to the discovery GWAS). Utilizing smaller numbers of samples in the replication stage may have resulted in reduced power to validate each of the identified loci in individual replication datasets. Nevertheless, all genome-wide hits at the discovery stage were directionally consistent and showed statistical significance in the replication set of SCHS population samples. Another potential limitation of the study may be the use of relative average LTL measurements determined through higher throughput qPCR approaches in our Singaporean Chinese GWAS datasets and this may be less reliable than other procedures like Southern Blot. However, the latter utilizes much higher amounts of DNA, is more laborious and may be unsuitable for large-scale studies[55–57]. Nevertheless, studies have indicated strong correlation of LTL measured through various approaches and the relatively large-sample sizes utilized in our GWAS study may have, to a certain extent, offset potential measurement errors due to the qPCR approach. Furthermore, the robust validations of multiple loci previously identified for LTL and the identification of loci with strong relevance to telomere biology provide a degree of confidence in our study findings.

In conclusion, our study utilizing East-Asian subjects and trans-ethnic meta-analysis procedures, expands on and provides biological insights into the genetic control of human telomere length homeostasis. We additionally identified an association between LTL and a LTL-associated variant (rs7253490) with respiratory infection deaths that may indicate a potential role of LTL in human immune competence.

## Methods

**Study datasets.** The SCHS is a prospective cohort of 63,257 Singaporean Chinese participants who were recruited between April 1993 and December 1998. Detailed methodology for the SCHS including prior telomere length measurement methodology have been previously described[11,58–60]. The study population was constituted by the two major dialect groups of Southern Han Chinese in Singapore, the Hokkiens and the Cantonese. At recruitment, subjects were interviewed in-person using a structured questionnaire which included information on demographics, body weight and height, lifetime use of tobacco, alcohol consumption, menstrual/reproductive history (women only), occupational exposure, medical history, and family history of cancer. Information on current diet and consumption of beverages was assessed via a 165 item food frequency questionnaire that had been validated against a series of 24 h dietary recall interviews and selected biomarker studies conducted on random subsets of cohort participants[60]. Beginning in 2000, we extended blood and urine collection to all surviving cohort participants. Of the 52,322 eligible subjects, 28,346 subjects (54% of eligible) donated blood samples. A total of 25,273 SCHS samples were whole genome genotyped using the Illumina Global Screening Array v1.0 ($N = 18,114$) and Illumina Global Screening Array v2.0 ($N = 7159$) and were utilized in the study for the discovery and replication GWAS stages, respectively. Informed consent was obtained from study participants and the present study was approved by the Institutional Review Boards of the National University of Singapore and the University of Pittsburgh. Main clinical variables for the SCHS study is provided in Supplementary Table 6.

The Singapore Study of Macro-angiopathy and Micro-vascular Reactivity in Type 2 Diabetes (SMART2D) dataset is a cross-sectional study conducted between August 2011 and February 2014 including a total of 2057 adults aged 21–90 years with T2DM. Detailed methodology including prior telomere length measurement methodology has been previously described[61] and for this study, 978 Singaporean Chinese samples from the SMART2D genotyped on the Illumina Humanomniexpress-24 Bead Chip were utilized. The other type 2 diabetic dataset, Diabetic Nephropathy (DN), is an ongoing study conducted from 2002 at Khoo Teck Puat Hospital, Singapore including 1984 adults with T2DM and detailed methodology has been previously described[62]. For this study, 624 Singaporean Chinese samples genotyped on the Illumina HumanOmniZhonghua Bead Chip were utilized. Individual written informed consent was obtained prior to enrollment in these studies and Singapore National Health Group domain-specific ethics approval were obtained.

**Telomere length measurements.** QIAamp DNA Blood kits (Qiagen, Valencia, CA) were used to extract DNA from peripheral blood in the SCHS study samples. Relative telomere length was determined using monochrome multiplex quantitative PCR (qPCR)[63]. Telomere length for each study sample was determined as the ratio of telomere (T) and albumin (S) gene copy numbers, relative to a reference sample. DNA sample for the standard curve in the study was generated using 77 SCHS samples with equimolar concentration. Relative telomere lengths of these 77 samples were within 10% of the population mean. Pooled DNA from these samples was used as the reference sample in concentrations of 4, 0.8, 0.16, and 0.032 ng/μl with 8 replicates each. Thermal cycling was performed with the Applied Biosystem 7900 HT, using PCR cycling conditions for telomere length determination. The PCR parameters were 15 min at 95 °C (Stage 1), 2 cycles of 15 s at 94 °C, 15 s at 49 °C (Stage 2), and 32 cycles of 15 s at 94 °C, 10 s at 62 °C, 15 s at 74 °C with signal acquisition, 10 s at 84 °C and 15 s at 88 °C with signal acquisition (Stage 3)[63]. 384-well plate-based normalization was performed on relative telomere length (T/S ratio) of study samples[63]. All qPCR experiments were performed in duplicates and the average T/S ratio value was used for subsequent analysis. Mean coefficient of variation of duplicates for telomere length in the study was 3.5%.

A similar quantitative qPCR methodology was utilized to measure telomere length from peripheral blood DNA from the SMART2D and DN subjects[63,64]. Relative telomere length (T/S ratio) for each study sample was determined as the ratio of telomere (T) and b-globin (S) gene copy numbers. DNA for the standard curve in the study was generated using commercially available human G304A DNA. This reference DNA was diluted to produce 6 concentrations from 8 to 0.25 ng/μl and ran in duplicates in each qPCR plate. We further utilized HepG2 DNA as an experimental control in each qPCR plate and the average normalizing factor was used to adjust T/S ratio values for each subject. All T/S ratio values for the SMART2D and DN subjects were obtained in duplicates, and the average of the two values were used for subsequent analyses. The mean coefficient of variation of duplicates for telomere length in the study samples was below 1.8%.

**Mortality assessment.** In the SCHS dataset, all-cause, cardiovascular, respiratory, and cancer deaths from the date of the baseline interview through 31 December 2017 were identified through linkage with the nationwide registry of births and deaths in Singapore. The International Classification of Diseases (ICD) 9th (ICD9)[65] or 10th (ICD10)[66,67] revision codes were used to classify causes of deaths from cardiovascular diseases [ICD9 (390–459) or ICD10 (I00–I99)], respiratory diseases including pneumonia and influenza [ICD9 (480–488) or ICD10 (J09–J18)], and COPD [ICD9 (490–496) or ICD10 (J40–47)] and cancer [ICD9 (140–208) or ICD10 (C00–C97)].

**Genotyping and imputation.** 18,114 SCHS samples were genotyped on the Illumina Global Screening Array v1.0 and used as the discovery dataset in the study. 7159 SCHS samples were genotyped on the Illumina Global Screening Array v2.0 and utilized in the replication stage. Quality control (QC) procedures of samples are detailed in Supplementary Table 8. Briefly, samples with call-rate <95.0% ($N = 176$) and extremes in heterozygosity (> or <3 SD, $N = 236$). Identity-by-state measures were performed by pair-wise comparison of samples to detect 1st and 2nd degree related samples and one sample, with the lower call-rate, from each relationship was excluded from further analysis ($N = 1625$). Principal component analysis (PCA) together with 1000 Genomes Projects reference populations and within the SCHS samples were performed to identify possible outliers from reported ethnicity and 55 samples were excluded. Samples that passed GWAS QC procedures were observed to cluster tightly with PCA analyses (Supplementary Fig. 14). GWAS genotyping and QC procedures for the SMART2D and DN datasets have been described previously[61,62]. Raw LTL data from each dataset were normalized by rank-based inverse normalization (z-scores) and the samples with extreme levels of Z-LTL (>3 or <−3 SD, $N = 75$) were excluded. 16,759 SCHS samples genotyped on the Illumina Global Screening Array v1.0 and 6,337 SCHS samples genotyped on the Illumina Global Screening Array v2.0 passed QC procedures and were available for subsequent statistical analysis and utilized for the discovery and replication stages of the Singaporean Chinese GWAS study (Supplementary Table 8).

For SNP QC (Supplementary Table 9), sex-linked and mitochondrial SNPs were removed, together with gross Hardy–Weinberg equilibrium (HWE) outliers ($P < 1 \times 10^{-6}$) were excluded in each dataset used in the study. SNPs that were monomorphic or with a minor allele frequency (MAF) < 1.0% and SNPs with low call-rates (<95.0%) were excluded. We imputed for additional autosomal SNPs with IMPUTE v2 using the cosmopolitan 1000 Genomes haplotypes as reference panel (Phase 3). SNPs with impute information score < 0.8, MAF < 1.0%, HWE $P < 1 \times 10^{-6}$ as well as non-biallelic SNPs were excluded from subsequent analyses. Alleles for all SNPs were coded to the forward strand and mapped to HG19. In total, 6,407,959 genotyped and imputed SNPs were available for statistical analyses after QC procedures for the SCHS discovery GWAS study and 6,406,238 genotyped and imputed SNPs were available for statistical analyses after QC procedures for the SCHS replication GWAS study.

**Statistical analysis**. Linear regression analysis for associations with normalized LTL (Z-LTL) was run in each dataset separately with the genome-wide association toolset SNPTEST (v2). All GWAS regression analyses were adjusted for age, sex and the top three principal components of population stratification (PCs 1–3). In each linear regression analysis, the score function was utilized to enable incorporation of imputation dosages in the regression model. Genomic inflation factor ($\lambda$) of association results were used to evaluate levels of inflation of study results and these were determined to be marginal in individual datasets ($\lambda$ between 1.043 and 0.9874, Supplementary Table 9). Subsequently, association summary statistics from discovery and replication stages were combined using the inverse variance-weighted meta-analysis, assuming a fixed effects model to derive overall association values using the META (v1.5) toolset. Heterogeneity of effects in meta-analyzed data was determined using Cochran's Q and a Cochran's Q P-value (P_het) < 0.05 was determined to be significantly heterogeneous. For the 16 significantly associated SNPs in our study, we estimated the proportion of the Z-LTL variance explained by evaluating changes in adjusted $r^2$ values of regression models before and after inclusion of these SNPs into the model using the discovery, replication and combined set of SCHS samples. Weighted genetic risk score (wGRS) using all 16 Z-LTL associated SNPs was constructed in the SCHS study. We multiplied the number of risk alleles (telomere length reducing allele) at each Z-LTL associated SNP by their meta-analyzed effect estimates (Table 1)[68]. The weighted GRS were summed over all Z-LTL associated SNPs, and divided by the absolute average effect estimate of the 16 SNPs[68]. Cox proportional hazards regression was used to assess the association between Z-LTL, wGRS and individual SNPs with all-cause, cardiovascular, respiratory, and cancer mortality. Person-years were calculated for each study participant from the date of the baseline interview to the date of death, date of loss to follow-up, or 31 December 2017, whichever occurred first. These associations were adjusted for age, sex, population substructure (PC1–PC3), BMI, and smoking status (never-smokers vs current/ex-smokers). Mediation analyses were performed using the SEM module in Stata (ver15) to identify total effect of SNP on outcome variable, direct effect of SNP on outcome variable and indirect effect of SNP on outcome variable through mediating variable[69]. Subsequently, the proportion of SNP effects mediated was calculated as the proportion of indirect effect over the total effect. All mediation analysis adjusted for age, sex, and population substructure (PC1–PC3).

**Functional annotation of SNPs**. Identified lead SNPs in the study were functionally annotated using the SNP2GENE function in Functional Mapping and Annotation (FUMA)[26]. All SNPs in LD ($r^2 > 0.6$ in 1000G ASN panel) with lead signals were identified and potential functionality evaluated through CADD and other bioinformatics toolsets (i.e., SIFT and Polyphen2). Regional genes (within 200 kb of lead SNP) at identified loci were mapped (LTL regional genes) and annotated using the GENE2FUNC function in FUMA. QTL assessment using all regional genes was performed using circulating immune cell-type specific expression QTL (eQTL), methylation QTL (mQTL), and histone modification QTL (hQTL) data from the BLUEPRINT (BLUEPRINT of Haematopoietic Epigenomes) and DICE (database of immune cell expression, expression quantitative trait loci and epigenomics) studies[29,30]. Subsequently, LD between QTL SNPs and GWAS lead SNPs were evaluated (1000 genomes ASN panel). For identified QTLs that were in LD with GWAS lead SNPs ($r^2 > 0.60$, 1000 genomes ASN panel) co-localization analysis was performed to determine the probability of a shared common causal variant for both QTL and LTL signals[70]. Analysis was performed by extracting summary association results for all significant regional QTL SNPs ($P_{Adj} < 5 \times 10^{-8}$, 2 Mb region) from the BLUEPRINT database (http://blueprint-dev.bioinfo.cnio.es/WP10/)[29] and their corresponding LTL GWAS association results from the SCHS meta-analysis and evaluated using the coloc.abf R package, using default priors (prior probability that a SNP is associated with first trait, p1 = $1 \times 10^{-4}$, prior probability that a SNP is associated with second trait, p2 = $1 \times 10^{-4}$, and prior probability that a trait is associated with both traits, p12 = $1 \times 10^{-5}$).

Gene set enrichments in GO terms and previous GWAS catalogs were performed for LTL regional genes. Implicated genes from GWAS studies were obtained from the Mapped genes/column from GWAS catalog (https://www.ebi.ac.uk/gwas/home) through FUMA analysis (http://fuma.ctglab.nl/)[26]. In GWAS catalog, if the reported SNP from previous GWAS studies is located within a gene, that particular gene would be listed under the Mapped gene/s and if the cataloged

SNP is intergenic, the nearest upstream and downstream genes would be listed. In gene-set enrichments, FUMA analysis tests for over-representation of LTL regional genes using hypergeometric tests[26]. The set of background genes (i.e., the genes against which the set of LTL regional genes are tested against) is 19,264 protein-coding genes and enrichment P-values were adjusted using Benjamini–Hochberg correction for multiple tests[26]. Enrichments of regional LTL genes in previous genes implicated in gene expression (1497 genes) and methylation (353 genes) clocks were also evaluated[31,32]. Additional canonical pathway enrichments were performed for identified mapped genes using Ingenuity Pathway Analysis (IPA) version 01-10.

## Data availability

Singapore Chinese LTL GWAS meta-analysis data for all SNPs evaluated in the study is available in https://doi.org/10.6084/m9.figshare.8066999. ENGAGE consortium LTL GWAS meta-analysis data is available in https://downloads.lcbru.le.ac.uk/engage. BLUEPRINT epigenome data is available in http://blueprint-dev.bioinfo.cnio.es/WP10. DICE epigenome data is available in https://dice-database.org/.

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

## Acknowledgements

The Singapore Chinese Health Study was supported by grants from the National Medical Research Council, Singapore (NMRC/CIRG/1456/2016), and the National Institutes of Health (R01 CA144034 and UM1 CA182876). The Singapore Study of Macro-angiopathy and Micro-vascular Reactivity in Type 2 Diabetes (SMART2D) cohort was supported by grants from the National Medical Research Council, Singapore (NMRC/PPG/AH(KTPH)/2011 and NMRC/CIRG/1398/2014). The Diabetic Nephropathy (DN) cohort was supported by grants from Alexandra Health Fund Private Limited (SIG II/15205). Telomere studies in the SMART2D and DN cohorts were supported by Alexandra Health Enabling Grants (AHEG1622 and AHEG1714). S.-C. Lim and W.-P. Koh were supported by National Medical Research Council, Singapore (NMRC/CSA-INV/0020/2017 and NMRC/CSA/0055/2013, respectively). C.C. Khor was supported by National Research Foundation Singapore (NRF-NRFI2018-01).

## Author contributions

R.D., W.P.K., J.-M.Y., C.C.K. and C.-K.H. contributed to the study design. S.L., S.C.L., J.-M.Y. and W.P.K. contributed to the recruitment, sample collection, and data processing. K.B.B., J.A.-H., R.W., M.Y. and R.L.G. generated telomere length data. R.D., Z.L., L.W., W.Y.M., K.S.S. and C.C.K. generated genotyping data. R.D., X.C., R.L.G., Y.F., J.L., R.M.v.D., C.C.K. and C.-K.H. contributed to the statistical and bioinformatics analyses. R.D., C.C.K., and C.-K.H. drafted the manuscript. All authors critically reviewed the manuscript.

## Additional information

**Competing interests:** The authors declare no competing interests.

