## [Peer Review File · Nature Communications]

Reviewers' comments:

Reviewer #1 (Remarks to the Author):

This manuscript describes a large GWAS of leukocyte TL conducted in a Singaporean Chinese population. This manuscript identifies several known associations, several novel loci, and several independent associations at known loci. The authors also report associations of TL SNPs (and SNP score) with various health outcomes, including strong associations with respiratory disease mortality. This is an important contribution to our knowledge of genetic determinants of TL because this study is one of the largest GWAS of TL to date and is conducted in a population of non-European ancestry. My comments and suggestions are below:

Results:

--Please report association of TL with age and sex. including variation in TL explain by each.

--The inflation factor is reported, but please mention here any steps taken to reduce inflation, such as PCs

--Fig 1. Label each peak with a gene or region name that will be recognizable to readers familiar with the TL GWAS literature.

--Table 1. rather than "independent SNP association", how about "known locus, independent SNP"

--Table 2-3. Some Mendelian randomization analyses (based on summary statistics) would be helpful here. such as making a plot of the SNPs association with TL vs assoc with disease phenotypes. Also some tests of the MR assumptions using MR-Egger, etc.

--Define FUMA on first use.

--"eQTL analysis based on GTEx tissue expression data indicated that 10 of the lead SNPs ... were significant cis eQTLs (FDR < 5 x 10⁻⁸) for regional genes in multiple tissues (Supplementary table 5)." But did these SNPs co-localize with any eQTL(s), suggesting that the causal variant for TL is also the causal variant for gene expression? this point applies to the discussion section as well. if there's no evidence for co-localization, then the effects of these SNPs are likely not through the eQTLs observed in GTEx.

--"Gene Ontology (GO) term analyses and canonical pathway analyses based on the 268 mapped regional genes indicated significant enrichments in pathways associated with telomere and DNA repair (Supplementary figures 9 – 11)." Please state what these are here in the text.

--"GO cellular component analyses also indicated that specific molecules identified from the study were implicated at different subcellular regions in the cells (for eg. OBFC1, TINF2, TERT, POT1 and TERF1 were components of the telomere cap complex, Supplementary figure 10A)." its nice to see

this, but fairly obvious based on the gene names reported earlier in the manuscript. Anything else interesting to point out here about GO the results?

--PARP1 was reported by Delgado et al 2018, as a "suggestive signal". is your PARP1 SNP in strong LD with the SNP reported by Delgado et al?

--Figs 2, 3, 4 might be better (and easier to see/read) if the association plots were stacked vertically. and you would not have to duplicate the genes underneath each association plot.

--fig 5: exactly how was a gene assigned to prior GWAS signals?

--Fig 5: do these results hold up if you remove the genes that come from the regions that have a strong candidate gene (i.e., a telomere-related gene?). For most of these regions that contain a telomere-related gene, those additional "regional genes" should be irrelevant, right?

--you report an R2 of just over 4%. Is this a model with all 19 SNPs as predictors? What is the adjusted R2? That might be more appropriate in this case. And is this from the discovery dataset or replication data set or both?

Discussion:

---"Over-representation of mapped loci especially in the ovary, may indicate the importance of telomere regulation in germ cells for increased cell proliferative capacity." I don't think you can claim this. It is my understanding that samples of ovary tissues (such as those used in GTEx) will not contain a substantial amount of germ cells (rather, other cell types). In contrast, testis samples will contain a large component of germ cells (or "pre-germ cells").

--Sample size is always an issue for power of Mendelian randomization. So please comment on the direction of the MR estimates are consistent with previous studies and your associations with measured LTL.

--also, if new MR analyses are explored, are results consistent with a causal impact of TL on these traits? or are there outlier SNPs that appear to violate assumptions and drive the observed associations for the allele score?

Methods:

-- "Detailed methodology for the SCHS including prior telomere length measurement methodology have been previously described [62-64]." These references 62-64 do not appear to be studies of TL. are these the correct references?

--additional details on the statistical methods underlying the various enrichment analyses are needed.

Reviewer #2 (Remarks to the Author):

Dorajoo and colleagues has completed a large GWAS study of LTL in Singaporean East-Asian populations with discovery and replication, as well trans-ethnic meta-analysis/consistency of signals with the European ancestry ENGAGE study. They have discovered and replicated several novel LTL loci with plausible explanatory genes at most loci. This is a significant step forward in LTL genetics in general, and in addressing associated variation in additional ancestry populations. The LTL trait is of interest in multiple diseases and in aging in general.

Principle findings:

- 8 new loci and 3 independent associations at known loci for LTL
- 16 genome-wide loci explain ~ 4.05% of phenotypic variation in LTL among SCHS study
- LTL variant rs7253490 and respiratory infection deaths
 - o Corroborates a potential role of LTL in human immune competence

Major Comments

1. The functional annotation and gene set enrichment analyses are not particularly informative (perhaps more confirmatory) and it is not clear that they do add to our understanding of telomere biology

- lines 192/193, What is the motivation of reporting mapped regional genes show significant upregulations especially in ovarian tissue?

- why not do a summary statistics data-based analysis of larger (than GTex) leukocyte eQTL datasets from the literature?

- leukocyte epigenetic regulatory datasets are abundant as well and would be worthwhile to intersect with peak SNPs or those in tight LD

2. Association of LTL with mortality due to chronic diseases is well established. Is anything being reported for the first time in Singaporean East-Asians?

- How much of the association between shorter LTL and death from respiratory infection is explained by the rs7253490 SNP?
- Given higher smoking rates generally in East Asian populations, is this effect stronger than prior studies potentially due to smoking interactions (do you have former/current smoker status to examine such a question?)
- Please elaborate how the chr19 ZNF gene locus could play a role in immune cell competence.
- Does the chr19 lead SNP rs7253490 perform better than the previously reported index SNP rs8105767 in cox proportional hazards analysis of respiratory mortalities?

3. LTL is important to aging processes and age-related diseases. In recent years, it has also been determined that there are epigenetic/methylation clocks (Horvath et al.; multiple papers) and gene expression clocks (Peters et al., Nat Commun) that relate to individuals who are more rapid or slower "agers". These clocks appear somewhat orthogonal and may have different and complementary predictive capacities, and may intersect in some cases with LTL biological loci. It may be informative to intersect/annotate your known and novel loci against these methylation-age and expression-age associated gene loci lists and highlight notable intersections (perhaps in Supplementary table and Discussion)

Minor Comments

1) Introduction - a heritability that can be "as high as 80%" seems a potential over-estimate. One study in n=383 supported that with a 12% SE. Another twin study n=115 found 78% heritability. However, these studies may not have adequately accounted for batch and environmental effects in measures and may have over-estimated the heritability. Other studies (and larger studies) had lower estimates: e.g., Lee et al. 2013: 50-60%, Andrew et al. 2006: 36% [18-48%], Hjelmborg 2015: 64% at baseline; 28% heritability of LTL attrition rate.

As shown by Andrew et al. the higher results for example of Slagboom et al. 1994 likely had to do with not accounting for environmental variable effects on the measurements. I suggest giving a range (rather than a maximum for heritability), or a more conservative interpretation of moderately heritable.

2) Overall the GWAS LTL references are well cited. You could consider adding Zeiger et al. 2018 Sci Rep PMID 30185882. This may also be helpful in terms of the underlying point that diverse population studies are needed and motivate the current study (ref 4;33-34 cited in Intro)

- 3) Confidence Intervals should be given in the text whenever Hazards Ratios are given
- 4) Are the coding SNPs in PARP1 and DCAF4 with high CADD and deleteriousness scores in functional protein domains?
- 5) Figure 1 Manhattan - locus labels would be helpful to reader (and lambda value)
- 6) LocusZoom plots appear grainy and stretched in aspect ratio
- 7) Capitalize "supplementary table" in several places for consistency
- 8) You should make it clear in main text that replication is not an entirely independent dataset, rather a subset of SCHS participants
- 9) Table 1 will obviously need to be formatted to fit a page (some boxes currently cutoff such as gene names for chr19 region)
- 10) lines 119/120, "twelve loci surpassing LTL association" but there are 11 loci in Table 1
- 11) lines 121/122, the 7 genome-wide significant signals from discovery would be replicated after adjusting for 7 tests, but not for 12 as reported in Table 1
- 12) lines 164/165, should Figure 4 be showing the trans-ethnic meta-analysis for DCAF4 gene locus or the SCHS meta-analysis?
- 13) LTL GWAS effect allele in TableS5 (GTEx) is not same as Test allele in Table 1 reducing C allele of rs7253490 would be associated with decreased expression of ZNF257 in lung

We thank the editor and the reviewers for thoroughly examining our manuscript and for their comments and suggestions. The points raised have helped to improve the data presented in our manuscript. We have provided our responses to all questions raised by reviewers below and have indicated on changes made to the revised manuscript.

Reviewers' comments:

Reviewer #1 (Remarks to the Author):

This manuscript describes a large GWAS of leukocyte TL conducted in a Singaporean Chinese population. This manuscript identifies several known associations, several novel loci, and several independent associations at known loci. The authors also report associations of TL SNPs (and SNP score) with various health outcomes, including strong associations with respiratory disease mortality. This is an important contribution to our knowledge of genetic determinants of TL because this study is one of the largest GWAS of TL to date and is conducted in a population of non-European ancestry. My comments and suggestions are below:

Results:

--Please report association of TL with age and sex. including variation in TL explain by each.

Response: We thank the reviewer for this suggestion. We performed the analysis as suggested and observed strong association between TL with age ($P=1.03 \times 10^{-126}$; variance explained = 6.3%) and sex ($P=2.57 \times 10^{-51}$; variance explained = 1.29%). We have included these data on page 7 of the revised manuscript and presented them in supplementary table 3.

“Increased age and males were highly associated with reduced LTL in our dataset, explaining approximately 6.30% and 1.29% of phenotypic variation (Supplementary table 3).” (Page 7 in revised manuscript).

--The inflation factor is reported, but please mention here any steps taken to reduce inflation, such as PCs.

Response: To reduce inflation, the top three principal components (PC1 – PC3) were included as covariates in the GWAS association analyses. These are reflected in the Methods section on page 21 of the revised manuscript.

“All GWAS regression analyses were adjusted for age, sex and the top three principal components of population stratification (PCs 1-3).” (Page 21 in revised manuscript).

--Fig 1. Label each peak with a gene or region name that will be recognizable to readers familiar with the TL GWAS literature.

Response: We thank the reviewer for this suggestion and have included genes relevant for TL for each genome-wide significant peak on Fig 1.

--Table 1. rather than “independent SNP association”, how about “known locus, independent SNP”

Response: We have modified Table 1 to reflect this.

--Table 2-3. Some Mendelian randomization analyses (based on summary statistics) would be helpful here. such as making a plot of the SNPs association with TL vs assoc with disease phenotypes. Also some tests of the MR assumptions using MR-Egger, etc.

Response: While measured LTL was significantly associated with various mortalities in our study (Table 2), our genetic data (combined wGRS and individual SNPs) did not show significant associations with disease mortalities, except for the rs7253490 SNP (*ZNF257*) with respiratory disease mortality (Table 3; main text page 11). As the reviewer suggests (below), the smaller sample sizes used in this study as compared to recent large-scale Mendelian randomization studies for LTL [ref:14] may have resulted in reduced power to detect robust significant associations between LTL risk SNPs and mortalities.

At the rs7253490 locus where we observed significant associations with respiratory infection deaths, we have now performed a mediation analysis (as suggested by Reviewer 2) and observed that this association with respiratory infection deaths was not strongly mediated through LTL (proportion of the rs7253490's effect on respiratory infection disease deaths mediated through LTL = 3.27%, Supplementary table 11 and Supplementary figure 13 in the revised manuscript). This suggests that rs7253490 is unsuitable, as a genetic proxy, for Mendelian randomization between LTL and respiratory infection death. We have therefore not carried out a formal Mendelian randomization analysis in the study and instead, have reported on the associations of measured LTL and genetic determinants of LTL with various disease mortalities in our Chinese study samples.

“Mediation analysis was performed to evaluate if the respiratory infection mortality effect of rs7253490 was mediated through LTL. Although the effect of rs7253490 on respiratory infection deaths mediated through LTL was significant ($P = 0.014$), the proportion mediated was modest (3.27%) (Supplementary table 11 and Supplementary figure 13).” (Page 11 in revised manuscript)

--Define FUMA on first use.

Response: We have now defined FUMA on first use.

“FUMA GWAS (Functional Mapping and Annotation of Genome Wide Association Studies) [36] ...” (Page 8 in the revised manuscript text).

--“eQTL analysis based on GTEx tissue expression data indicated that 10 of the lead SNPs ... were significant cis eQTLs ($FDR < 5 \times 10^{-8}$) for regional genes in multiple tissues (Supplementary table 5).” But did these SNPs co-localize with any eQTL(s), suggesting that the causal variant for TL is also the causal

variant for gene expression? this point applies to the discussion section as well. if there's no evidence for co-localization, then the effects of these SNPs are likely not through the eQTLs observed in GTEx.

Response: We appreciate the reviewer's point. To address the co-localization of GWAS SNPs and eQTLs, we have now replaced the data on eQTLs observed in GTEx with more biologically relevant circulating immune cell-type specific epigenomic data from BLUEPRINT (BLUEPRINT of Haematopoietic Epigenomes) [ref: 39] and DICE (database of immune cell expression, expression quantitative trait loci and epigenomics) [ref: 40] study groups. This is because the tissue of relevance for leukocyte telomere length (LTL) are the leukocytes themselves as well as other hematopoietic cells. At four loci, we identified co-localization of GWAS loci with QTL SNPs that were in strong LD (r^2 between 0.985 – 0.916, 1000G ASN panel). These analyses implicate *MPHOSPH6*, *POT1*, *ATM* and *ZNF257* as likely functional LTL genes at the four GWAS loci in specific immune cell-types. These data are provided in Supplementary table 7 and indicated in the results section on page 9 and in the discussion section on page 12-13.

“For the 268 regional LTL genes, 454 significant quantitative trait loci [expression QTL (eQTL), methylation QTL (mQTL) and histone modification QTL (hQTL), Bonferroni corrected $P < 5 \times 10^{-8}$)] SNPs were identified from the BLUEPRINT (BLUEPRINT of Haematopoietic Epigenomes) blood immune cell-type specific (monocytes, T cells and neutrophils) epigenome data [39] (Supplementary table 7). Four LTL GWAS lead SNPs from the study co-localized with four QTL SNPs (r^2 between 0.985 – 0.916, 1000G ASN panel). These include hQTL SNPs, rs2911429 and rs9969187, affecting histone peaks at *MPHOSPH6* in neutrophils and *POT1* in monocytes, respectively. The rs7253490 GWAS lead SNP co-localized with an eQTL SNP (rs17554725, $r^2 = 0.944$, 1000G ASN panel) that affects *ZNF257* gene expression in both monocytes and T cells. The A allele of rs17554725 was observed to decrease LTL levels in our data (Beta = -0.0396, Meta $P = 5.24 \times 10^{-5}$) and decreased *ZNF257* expression in both monocytes and T cells (Supplementary tables 1 and 7). The rs227080 GWAS lead SNP co-localized with another eQTL SNP (rs660429, $r^2 = 0.916$, 1000G ASN panel) that affects *ATM* expression in T cells. Rs660429 was also a significant eQTL in T cells for *ATM* expression in the DICE (database of immune cell expression, expression quantitative trait loci and epigenomics) study [40] (Supplementary table 8). The C allele of rs660429 was observed to decrease LTL levels (beta = -0.0526, Meta $P = 9.03 \times 10^{-9}$) in our data and increase *ATM* expression in T cells (Supplementary tables 1 and 7).” (Page 9 in the revised manuscript text)

“At four other LTL GWAS loci, identified lead SNPs co-localized with circulating immune cell-type QTL SNPs, potentially implicating *POT1*, *MPHOSPH6*, *ATM* and *ZNF257* as likely functional genes at these GWAS loci.” (page 12-13 in the revised manuscript text)

--“Gene Ontology (GO) term analyses and canonical pathway analyses based on the 268 mapped regional genes indicated significant enrichments in pathways associated with telomere and DNA repair (Supplementary figures 9 – 11).” Please state what these are here in the text.

Response: We have now listed these pathways in the manuscript text on page 9-10 of the revised manuscript.

“Gene Ontology (GO) term analyses and canonical pathway analyses based on the 268 mapped regional genes indicated significant enrichments in pathways associated with telomere and DNA repair such as 1)

telomere maintenance via telomere lengthening and telomerase activity, 2) base excision repair and 3) DNA double strand break repair (Supplementary figures 8 – 10).” (page 9-10 of the revised manuscript)

--“GO cellular component analyses also indicated that specific molecules identified from the study were implicated at different subcellular regions in the cells (for eg. OBFC1, TINF2, TERT, POT1 and TERF1 were components of the telomere cap complex, Supplementary figure 10A).” its nice to see this, but fairly obvious based on the gene names reported earlier in the manuscript. Anything else interesting to point out here about GO the results?

Response: We agree with the reviewer. Apart from insights linking OBFC1, TINF2, TERT, POT1 and TERF1 to the shelterin telomere cap complex, an additional finding from the GO cellular component analysis was that other molecules involved in DNA damage surveillance and repair mechanisms (such as PARP1, ATM, TOX4, THOC1 and SMCHD1) were not part of the cap complex and more localized to telomeric regions, emphasizing the importance of these DNA repair mechanisms in affecting cellular telomere lengths. We have now included these information in the results section (page 10) and enhanced the explanation on this point in the discussion section (page 13-14) in the revised manuscript text, as follows:

“... while *PARP1*, *ATM*, *TOX4*, *THOC1* and *SMCHD1* were associated with chromosome telomeric regions, Supplementary figure 9A).” (page 10)

“One of the known key function of shelterin is in preventing the activation of three DNA damage response enzymes ATM, ATR kinases and PARP1 [52-53]. It is noteworthy that variants in genes encoding for two of the three DNA damage response enzymes have been identified in this study to be associated with LTL (i.e *ATM* and *PARP1*).... Other regional genes identified in the study such as *TOX4*, *THOC1* and *SMCHD1* were also involved in DNA damage surveillance and repair mechanisms and were observed to localize at the telomeric region [55-57].” (page 13-14).

--PARP1 was reported by Delgado et al 2018, as a “suggestive signal”. is your PARP1 SNP in strong LD with the SNP reported by Delgado et al?

Response: The SNP downstream of *PARP1* reported as a suggestive signal by Delgado et al, 2018 (rs1151814) was in weak LD with our top hit within the *PARP1* gene (rs3219104) (pairwise $r^2 = 0.182$ in the Singapore Chinese). SNP rs1151814, reported by Delgado et al., was however nominally associated with LTL in our Singapore Chinese dataset (beta = -0.023, Meta P = 0.017, supplementary table 1). Conditioning the association at rs3219104 (which we observed in Singapore Chinese to exceed genome-wide significance at Meta P = 2.23×10^{-16} , Beta = -0.074) for rs1151814 did not result in an attenuation of the signal at rs3219104 (Meta P after conditional probability analysis = 9.18×10^{-15} , Beta = -0.075) suggesting that both *PARP1* markers could represent independent signals.

--Figs 2, 3, 4 might be better (and easier to see/read) if the association plots were stacked vertically. and you would not have to duplicate the genes underneath each association plot.

Response: We thank the reviewer for this suggestion and have stacked the association plots vertically for Figs 2, 3 and 4.

--fig 5: exactly how was a gene assigned to prior GWAS signals?

Response: For Figure 5, implicated genes from previous GWAS studies were obtained from the “Mapped gene/s” column from GWAS catalog (<https://www.ebi.ac.uk/gwas/home>) through the Functional Mapping and Annotation (FUMA) database [ref:36]. In GWAS catalog, if the reported SNP from previous GWAS studies is located within a gene, that particular gene would be listed under the “Mapped gene/s” and if the catalogued SNP is intergenic, the nearest upstream and downstream genes would be listed, as per-standard practice (<https://www.ebi.ac.uk/gwas/home>). We have now included these information in the methods section (page 23 of the revised manuscript).

“Implicated genes from GWAS studies were obtained from the “Mapped gene/s” column from GWAS catalog (<https://www.ebi.ac.uk/gwas/home>) through FUMA analysis (<http://fuma.ctglab.nl/>) [36]. In GWAS catalog, if the reported SNP from previous GWAS studies is located within a gene, that particular gene would be listed under the “Mapped gene/s” and if the catalogued SNP is intergenic, the nearest upstream and downstream genes would be listed.” (Page 23 of the revised manuscript).

--Fig 5: do these results hold up if you remove the genes that come from the regions that have a strong candidate gene (i.e., a telomere-related gene?). For most of these regions that contain a telomere-related gene, those additional “regional genes” should be irrelevant, right?

Response: We thank the reviewer for this suggesting this. When strong telomere-related genes (from table 1) were excluded from the gene-set analysis, the enrichment in genetic loci reported for telomere length was indeed abolished.

In addition, the observed significant enrichments in genetic loci reported for melanoma, glioma, non-glioblastoma glioma, glioblastoma, thyroid cancer, lung cancer, uterine fibrosis, interstitial lung disease, breast cancer in *BRACA1* mutation carriers, response to serotonin reuptake inhibitors and depression and BMI change over time were also abolished. This observation indicates that telomere-related genes could be also relevant in these diseases and traits. We have now included these new analyses in the results section (page 10) of the revised manuscript text. We have also added a figure analyzing gene-enrichment after excluding telomere-related genes in supplementary figure 11 of the revised manuscript.

“Repeating this analysis after exclusion of strong telomere-related genes (Table 1), expectedly abolished associations with loci implicated in telomere length genetic studies. Moreover, enrichments in gene loci reported for melanoma, glioma, non-glioblastoma glioma, glioblastoma, thyroid cancer, lung cancer, uterine fibrosis, interstitial lung disease, breast cancer in *BRACA1* mutation carriers, response to serotonin reuptake inhibitors and depression and BMI change over time were lost, perhaps indicating that telomere-related genes were also relevant in these diseases and traits (Supplementary figure 11).” (Page 10 in revised manuscript)

--you report an R² of just over 4%. Is this a model with all 19 SNPs as predictors? What is the adjusted R²? That might be more appropriate in this case. And is this from the discovery dataset or replication data set or both?

Response: The R² which we reported in the first submission was calculated by adding individual changes in adjusted R² for the 16 SNPs, using the combined set of SCHS samples. As recommended by the reviewer, we have now included all 16 genome-wide significant SNPs in the same regression model and calculated changes to adjusted R² with and without the inclusion of these 16 SNPs. These have been done separately in the discovery, replication and combined sets of SCHS samples. The changes to adjusted R² were 4.55%, 3.38% and 3.98% in the discovery, replication and combined SCHS samples, respectively. We have presented these data as supplementary table 4 and reported on the estimated proportion explained using the combined SCHS set in page 7-8 in the revised manuscript.

“In total the 16 genome-wide loci identified explained approximately 3.98% of the phenotypic variation in LTL among the samples from the combined SCHS study (Supplementary table 4), roughly doubling the variance explained by previous studies [27].” (Page 7-8 in revised manuscript).

Discussion:

----“Over-representation of mapped loci especially in the ovary, may indicate the importance of telomere regulation in germ cells for increased cell proliferative capacity.” I don’t think you can claim this. It is my understanding that samples of ovary tissues (such as those used in GTE_x) will not contain a substantial amount of germ cells (rather, other cell types). In contrast, testis samples will contain a large component of germ cells (or “pre-germ cells”).

Response: We thank the reviewer for indicating on this point and we have excluded this section from the revised manuscript text.

--Sample size is always an issue for power of Mendelian randomization. So please comment on the direction of the MR estimates are consistent with previous studies and your associations with measured LTL.

Response: We agree with the reviewer and it is likely that our sample sets with mortality information were underpowered to robustly demonstrate causal effects of LTL. Nevertheless, it is reassuring that at least the direction of effects between measured LTL (as well as the combined wGRS for LTL), and various mortalities were consistent with data from larger-scale Mendelian randomization studies (Tables 2 and 3). We have now stated this in the discussion section on page 14-15 of the revised manuscript.

“Previous Mendelian randomization studies have reported on a causal effect between longer LTL and increased risks of cancer and decreased risks of non-cancer disorders [14]. In this study, we report similar trends between measured LTL and deaths due to cancer, respiratory and cardiovascular diseases in the Singaporean Chinese population.” (page 14-15)

--also, if new MR analyses are explored, are results consistent with a causal impact of TL on these traits? or are there outlier SNPs that appear to violate assumptions and drive the observed associations for the allele score?

Response: The strongest disease association seen among the 16 SNPs showing genome-wide significant associations with LTL was between rs7253490 and respiratory disease deaths. As this was however only modestly mediated through LTL via mediation analysis, we have not performed any new Mendelian randomization evaluations.

Methods:

-- "Detailed methodology for the SCHS including prior telomere length measurement methodology have been previously described [62-64]." These references 62-64 do not appear to be studies of TL. are these the correct references?

Response: We thank the reviewer for highlighting this. The reference on telomere length measurement in the SCHS study is reference 19 - [19] Yuan JM, Beckman KB, Wang R, Bull C, Adams-Haduch J, Huang JY, Jin A, Opreko P, Newman AB, Zheng YL, Fenech M, Koh WP. Leukocyte telomere length in relation to risk of lung adenocarcinoma incidence: Findings from the Singapore Chinese Health Study. *Int J Cancer*. 2018 Jun 1;142(11):2234-2243]. We have rectified this in the manuscript on page 17 of the revised manuscript.

--additional details on the statistical methods underlying the various enrichment analyses are needed.

Response: We have now included additional details on the FUMA gene-set enrichment as well as the QTL assessment in the methods section of the revised manuscript on page 23 of the revised manuscript.

"QTL assessment using all regional genes was performed using circulating immune cell-type specific expression QTL (eQTL), methylation QTL (mQTL) and histone modification QTL (hQTL) data from the BLUEPRINT (BLUEPRINT of Haematopoietic Epigenomes) and DICE (database of immune cell expression, expression quantitative trait loci and epigenomics) studies [39-40]. Subsequently, LD between QTL SNPs and GWAS lead SNPs were evaluated (1000 genomes ASN panel). Gene set enrichments in gene ontology (GO) terms and previous GWAS catalogues were performed for LTL regional genes. Implicated genes from GWAS studies were obtained from the "Mapped gene/s" column from GWAS catalog (<https://www.ebi.ac.uk/gwas/home>) through FUMA analysis (<http://fuma.ctglab.nl/>) [36]. In GWAS catalog, if the reported SNP from previous GWAS studies is located within a gene, that particular gene would be listed under the "Mapped gene/s" and if the catalogued SNP is intergenic, the nearest upstream and downstream genes would be listed. In gene-set enrichments, FUMA analysis tests for over-representation of LTL regional genes using hypergeometric tests [36]. The set of background genes (i.e. the genes against which the set of LTL regional genes are tested against) is 19,264 protein-coding genes and enrichment p-values were adjusted using Benjamini-Hochberg correction for multiple tests [36]. Enrichments of regional LTL genes in previous genes implicated in gene expression (1497 genes) and methylation (353 genes) clocks were also evaluated [41-42]." (page 23 of revised manuscript)

Reviewer #2 (Remarks to the Author):

Dorajoo and colleagues has completed a large GWAS study of LTL in Singaporean East-Asian populations with discovery and replication, as well trans-ethnic meta-analysis/consistency of signals with the European ancestry ENGAGE study. They have discovered and replicated several novel LTL loci with plausible explanatory genes at most loci. This is a significant step forward in LTL genetics in general, and in addressing associated variation in additional ancestry populations. The LTL trait is of interest in multiple diseases and in aging in general.

Principle findings:

- 8 new loci and 3 independent associations at known loci for LTL
 - 16 genome-wide loci explain $\sim 4.05\%$ of phenotypic variation in LTL among SCHS study
 - LTL variant rs7253490 and respiratory infection deaths
- o Corroborates a potential role of LTL in human immune competence

Major Comments

1. The functional annotation and gene set enrichment analyses are not particularly informative (perhaps more confirmatory) and it is not clear that they do add to our understanding of telomere biology

- lines 192/193, What is the motivation of reporting mapped regional genes show significant upregulations especially in ovarian tissue?

Response: The motivation in this analysis using GTEx gene expression data was to evaluate if LTL implicated genes were enriched in any particular tissue, thus suggesting tissue-specific biological insights. However, as also indicated by Reviewer 1, the potential heterogeneity of gene expression data in various cell-types of tissues may actually limit the usefulness of such evaluations. In this light, we have excluded the GTEx analysis from the study. Instead, as recommended by the reviewer, we have replaced this with a more specific evaluation using immune cell-type specific epigenomic datasets (do see below) from the BLUEPRINT and DICE studies

- why not do a summary statistics data-based analysis of larger (than GTEx) leukocyte eQTL datasets from the literature?

Response: We agree with the reviewer. We have now evaluated the overlap between our GWAS hits, correlated list of SNPs, regional LTL gene data and circulating immune cell-type specific expression QTL (eQTL), methylation QTL (mQTL) and histone modification QTL (hQTL) data from the BLUEPRINT (BLUEPRINT of Haematopoietic Epigenomes) and DICE (database of immune cell expression, expression

quantitative trait loci and epigenomics) studies. At four of the identified LTL GWAS loci, we observed significant QTL SNPs that co-localized with GWAS lead SNPs (r^2 between 0.985 – 0.916, 1000G ASN panel). These analyses potentially implicate *MPHOSPH6*, *POT1*, *ATM* and *ZNF257* as likely functional LTL genes at these four GWAS loci in specific immune cell-types. These data are provided in Supplementary tables 7-8 and indicated in the results section on page 9 and in the discussion section on page 12-13 of the revised manuscript.

“For the 268 regional LTL genes, 454 significant quantitative trait loci [expression QTL (eQTL), methylation QTL (mQTL) and histone modification QTL (hQTL), Bonferroni corrected $P < 5 \times 10^{-8}$] SNPs were identified from the BLUEPRINT (BLUEPRINT of Haematopoietic Epigenomes) blood immune cell-type specific (monocytes, T cells and neutrophils) epigenome data [39] (Supplementary table 7). Four LTL GWAS lead SNPs from the study co-localized with four QTL SNPs (r^2 between 0.985 – 0.916, 1000G ASN panel). These include hQTL SNPs, rs2911429 and rs9969187, affecting histone peaks at *MPHOSPH6* in neutrophils and *POT1* in monocytes, respectively. The rs7253490 GWAS lead SNP co-localized with an eQTL SNP (rs17554725, $r^2 = 0.944$, 1000G ASN panel) that affects *ZNF257* gene expression in both monocytes and T cells. The A allele of rs17554725 was observed to decrease LTL levels in our data (Beta = -0.0396, Meta $P = 5.24 \times 10^{-5}$) and decreased *ZNF257* expression in both monocytes and T cells (Supplementary tables 1 and 7). The rs227080 GWAS lead SNP co-localized with another eQTL SNP (rs660429, $r^2 = 0.916$, 1000G ASN panel) that affects *ATM* expression in T cells. Rs660429 was also a significant eQTL in T cells for *ATM* expression in the DICE (database of immune cell expression, expression quantitative trait loci and epigenomics) study [40] (Supplementary table 8). The C allele of rs660429 was observed to decrease LTL levels (beta = -0.0526, Meta $P = 9.03 \times 10^{-9}$) in our data and increase *ATM* expression in T cells (Supplementary tables 1 and 7).” (Page 9 in the revised manuscript text)

“At four other LTL GWAS loci, identified lead SNPs co-localized with circulating immune cell-type QTL SNPs, potentially implicating *POT1*, *MPHOSPH6*, *ATM* and *ZNF257* as likely functional genes at these GWAS loci.” (page 12-13 in the revised manuscript text)

- leukocyte epigenetic regulatory datasets are abundant as well and would be worthwhile to intersect with peak SNPs or those in tight LD

Response: We thank the reviewer for this suggestion and as indicated above, this has now been evaluated as well in the revised manuscript.

2. Association of LTL with mortality due to chronic diseases is well established. Is anything being reported for the first time in Singaporean East-Asians?

Response: As pointed out by the reviewer, longer LTL has been associated with increased risks of cancers and shorter LTL with increased risks non-cancer diseases (coronary heart disease and COPD), similar to the trends we observed in our study. However, the association of reduced LTL with increased risks of respiratory infection mortality and the association between the LTL reducing SNP (rs7253490) with respiratory infection mortality in the Chinese population are reported for the first time here.

- How much of the association between shorter LTL and death from respiratory infection is explained by the rs7253490 SNP?

Response: We thank the reviewer for clarifying on this point and we have now explored these associations further. Adjusting the LTL-mediated respiratory infection mortality effects with the rs7253490 SNP did not attenuate the effect of LTL [HR = 0.867 (95%CI: 0.812 - 0.925) without adjustment and HR = 0.861 (95%CI: 0.812 - 0.925) after adjustment for rs7253490] and the rs7253490 SNP only explained a modest proportion of the respiratory infection effect (R^2 change = 0.08%). This may suggest separate effects of LTL and rs7253490 on respiratory disease, despite the genome-wide significant association between rs7253490 and LTL.

We further carried out a formal mediation analysis to evaluate if the effect of rs7253490 on respiratory infection deaths was mediated through LTL. Although the effect of rs7253490 on respiratory infection deaths mediated through LTL was significant ($P=0.014$), these analyses indicated that only a modest proportion of the rs7253490's effect was mediated through LTL (3.27%) and may indicate that rs7253490 may also exert its effect on respiratory diseases through additional mechanisms besides LTL. These data have now been included in the results, discussion and abstract of the manuscript (page 11 and 15 of revised manuscript) and presented in supplementary table 11 and supplementary figure 13.

“Mediation analysis was performed to evaluate if the respiratory infection mortality effect of rs7253490 was mediated through LTL. Although the effect of rs7253490 on respiratory infection disease deaths mediated through LTL was significant ($P = 0.014$), the proportion mediated was modest (3.27%) (Supplementary table 11 and Supplementary figure 13).” (page 11)

“At the same time however, it is important to note that much of the respiratory infection mortality effects for rs7253490 was not strongly mediated through LTL, signifying potential pleiotropic effects at this genetic locus.” (page 15)

“We further show that the LTL reducing SNP rs7253490 was also associated with respiratory infections ($P = 7.44 \times 10^{-4}$) however this effect may not be strongly mediated through LTL” (abstract)

- Given higher smoking rates generally in East Asian populations, is this effect stronger than prior studies potentially due to smoking interactions (do you have former/current smoker status to examine such a question?)

Response: We thank the reviewer for suggesting this analysis. Given the likely pleiotropic effect of rs7253490 we have now assessed if this SNP interacts with smoking status as well as with alcohol consumption, physical activity levels and obesity levels (i.e BMI) to affect respiratory infection associations. However, we do not identify any significant interactions. These data are included in supplementary table 13 and on page 15 of the revised manuscript.

“At the same time however, it is important to note that much of the respiratory infection mortality effects for rs7253490 was not strongly mediated through LTL, signifying potential pleiotropic effects at this genetic locus. Interaction analyses with multiple lifestyle factors (i.e smoking status, alcohol consumption, physical activity and obesity levels) however, did not identify interactions that modified the associations between rs7253490 and respiratory infection deaths (Supplementary table 13) in our study samples.” (page 15 of the revised manuscript)

- Please elaborate how the chr19 ZNF gene locus could play a role in immune cell competence.

Response: The eQTL evaluation in immune cells highlighted a SNP (in LD with the GWAS lead SNP at this locus) that reduced LTL also reduced *ZNF257* gene expression levels in T cells and monocytes. The literature surrounding *ZNF257* show limited characterization of the gene, save for a recent study suggesting that *ZNF257* may act as a transcription factor that affects downstream genes associated in immune response [ref: 63]. We have included these additional references on *ZNF257* in the discussion section on page 15 of the revised manuscript.

“Our eQTL look-up indicated that a LTL reducing SNP in LD with rs7253490 decreased *ZNF257* gene expression in T cells and monocytes, implicating *ZNF257* as a potential functional gene at this GWAS locus and *ZNF257* may function as a transcription factor that affects downstream genes linked to immune response [63].” (page 15 of the revised manuscript)

- Does the chr19 lead SNP rs7253490 perform better than the previously reported index SNP rs8105767 in cox proportional hazards analysis of respiratory mortalities?

Response: Yes indeed.

SNP rs7253490 [HR=1.181(1.084-1.285), $P=1.29 \times 10^{-4}$] has a marginally more significant association with respiratory mortalities than the rs8105767 SNP [HR=1.159 (1.080-1.264), $P=9.64 \times 10^{-4}$]. The 2 SNPs were in strong linkage disequilibrium in the 1000Genomes ASN panel ($r^2=0.8202$).

3. LTL is important to aging processes and age-related diseases. In recent years, it has also been determined that there are epigenetic/methylation clocks (Horvath et al.; multiple papers) and gene expression clocks (Peters et al., Nat Commun) that relate to individuals who are more rapid or slower "agers". These clocks appear somewhat orthogonal and may have different and complementary predictive capacities, and may intersect in some cases with LTL biological loci. It may be informative to intersect/annotate your known and novel loci against these methylation-age and expression-age associated gene loci lists and highlight notable intersections (perhaps in Supplementary table and Discussion)

Response: We thank the reviewer for this suggestion. 19 regional LTL genes (*ATM*, *DCAF4*, *PARP1*, *DDX10*, *ENOSF1*, *GSTO1*, *IPO4*, *IRF9*, *ITPRIP*, *LPCAT1*, *MARCH1*, *MSC*, *NPAT*, *RIPK3*, *RPSAP58*, *SH3PXD2A*, *SPTBN1*, *THOC1* and *ZNF800*) overlapped with loci from the gene expression clock (1497 genes). Pathway analysis based on these 19 overlapping genes also highlighted a significant enrichment in the DNA double-strand break repair by non-homologous end joining pathway (involving *ATM* and *PARP1*, $P_{Adj} = 0.013$) and reiterates the importance of these genes and DNA repair mechanisms in LTL homeostasis as well as in overall human ageing. Only 1 of the LTL regional genes overlapped with the 353 gene loci from the methylation ageing clock, suggesting that perhaps LTL related genes may not be particularly enriched for age-associated CpG methylation sites. However, without information on methylation data in our own study samples we were not able to formally evaluate on this. These data are provided in supplementary table 9 and have now been included in the results (page 10) and discussion sections of the revised manuscript (page 13-14).

“Additionally, we overlapped the 268 regional LTL genes with those previously reported for gene expression and methylation clocks of ageing [41-42]. 1 gene (*UCKL1*) overlapped with methylation-age associated gene loci [Supplementary table 9]. 19 LTL regional genes overlapped with expression-age associated gene loci [41] and indicated a significant enrichment in the DNA double-strand break repair by non-homologous end joining pathway (involving *ATM* and *PARP1*, $P_{Adj} = 0.013$) [Supplementary table 9 and Supplementary figure 12].” (page 10 of revised manuscript)

“Overlap of *ATM* and *PARP1* genes with gene loci previously identified as components of gene expression aging clocks [41], recapitulates the importance of DNA repair mechanisms (for eg. DNA double-strand break repair) in LTL homeostasis as well as in the overall human ageing process.” (page 13-14 of revised manuscript)

Minor Comments

1) Introduction - a heritability that can be "as high as 80%" seems a potential over-estimate. One study in $n=383$ supported that with a 12% SE. Another twin study $n=115$ found 78% heritability. However, these studies may not have adequately accounted for batch and environmental effects in measures and may have over-estimated the heritability. Other studies (and larger studies) had lower estimates: e.g., Lee et al. 2013: 50-60%, Andrew et al. 2006: 36% [18-48%], Hjelmborg 2015: 64% at baseline; 28% heritability of LTL attrition rate.

As shown by Andrew et al. the higher results for example of Slagboom et al. 1994 likely had to do with not accounting for environmental variable effects on the measurements. I suggest giving a range (rather than a maximum for heritability), or a more conservative interpretation of moderately heritable.

Response: We agree with the reviewer and have indicated that heritability of LTL is between 30%-60%.

“Heritability of LTL levels is approximately 30-60%....” (page 3 of the revised manuscript).

2) Overall the GWAS LTL references are well cited. You could consider adding Zeiger et al. 2018 Sci Rep PMID 30185882. This may also be helpful in terms of the underlying point that diverse population studies are needed and motivate the current study (ref 4;33-34 cited in Intro)

Response: We thank the reviewer for highlighting this recent paper and we have included this in the reference list – ref [35].

“.... it is likely that performing genetic studies in diverse populations could uncover additional genetic loci associated with LTL, as already seen in the South-Asian and African ancestry populations [32 and 35]....” (page 4 of revised manuscript).

3) Confidence Intervals should be given in the text whenever Hazards Ratios are given

Response: We have now included confidence intervals for Hazard ratios in the abstract and results sections (page 10-11 of the revised manuscript).

4) Are the coding SNPs in PARP1 and DCAF4 with high CADD and deleteriousness scores in functional protein domains?

Response: The coding SNP in *DCAF4* is close to the WD40 domain and the *PARP1* coding SNP affects the catalytic domain. These information have been included in the results section of the revised manuscript on page 8-9.

“However, strong Combined Annotation Dependent Depletion (CADD) annotations [37] (CADD > 20, top 1% of deleterious variants in the human genome) and potentially protein damaging predictions were identified at 2 SNPs that were in LD with GWAS lead SNPs ($r^2 > 0.99$, Supplementary table 6). These include the exonic SNP at the *DCAF4* gene locus (rs3815460, 345S>C) that was near a WD40 domain (UniProtKB) and another exonic SNP (rs1136410, 762V>A) at the *PARP1* gene locus (CADD = 32, top 0.1% of deleterious variants in the human genome) that has been reported to affect the catalytic domain of PARP1 through the binding of NAD⁺ at residue 762 [38].” (page 8-9 of the revised manuscript)

5) Figure 1 Manhattan - locus labels would be helpful to reader (and lambda value)

Response: We have now included locus labels for Figure 1 and indicated the lambda value.

6) LocusZoom plots appear grainy and stretched in aspect ratio

Response: LocusZoom plots on figure 2-4 have been enhanced.

7) Capitalize “supplementary table” in several places for consistency

Response: We thank the reviewer for highlighting this and have made the amendments.

8) You should make it clear in main text that replication is not an entirely independent dataset, rather a subset of SCHS participants

Response: We have now stated this specifically on page 5 of the revised manuscript.

“... were brought forward for replication in a subset of samples from the SCHS (6,337 additional Singaporean Chinese population samples, Table 1 and Supplementary table 1).” (page 5 of revised manuscript)

“All 7 genome-wide significant signals identified from the discovery GWAS showed consistent and significant replication in the replication samples (Table 1).” (page 5 of revised manuscript)

9) Table 1 will obviously need to be formatted to fit a page (some boxes currently cutoff such as gene names for chr19 region)

Response: We have now reformatted Table 1 to fit a page.

10) lines 119/120, "twelve loci surpassing LTL association" but there are 11 loci in Table 1

Response: We thank the reviewer for highlighting this and have rectified this on page 5 of the revised manuscript.

"Eleven loci surpassing LTL association $P < 1 \times 10^{-6}$ in the discovery stage analysis were brought forward for replication in a subset of samples from the SCHS (6,337 additional Singaporean Chinese population samples, Table 1 and Supplementary table 1)." (Page 5 of the revised manuscript)

11) lines 121/122, the 7 genome-wide significant signals from discovery would be replicated after adjusting for 7 tests, but not for 12 as reported in Table 1

Response: For added stringency we had adjusted the association levels in the SCHS replication data for total number of loci brought forward for replication (11 gene loci with Discovery GWAS p-value $< 1 \times 10^{-6}$). Nevertheless, all genome-wide SNPs reported in the study were observed to robustly replicate in either the SCHS replication sample set and/or the large-scale ENGAGE consortia dataset.

12) lines 164/165, should Figure 4 be showing the trans-ethnic meta-analysis for DCAF4 gene locus or the SCHS meta-analysis?

Response: We have now used trans-ethnic meta-analysis data to plot Figure 4.

13) LTL GWAS effect allele in TableS5 (GTEx) is not same as Test allele in Table 1 reducing C allele of rs7253490 would be associated with decreased expression of ZNF257 in lung.

Response: GTEx data has been replaced with QTL data from immune cells, which we agree with the reviewer would be more relevant for the study.

Editor's comments:

Please be aware that for certain types of new data, including most types of genetic data, journal policy is that deposition in a community-endorsed, public repository is generally mandatory prior to publication. Data submission can be a lengthy process, and we strongly suggest that you begin this well in advance of potential publication to avoid delays later on. Please include a statement about data availability in your point-by-point letter accompanying your revisions. Specifically, please state how GWAS summary statistics will be made available.

Please see decision letter above for more detailed information about data requirements and policy. If you are unable to make your data publically available for exceptional reasons, please get in touch with me now to discuss this further.

Response: We have provided GWAS summary data for all SNPs with metaP < 0.01 as part of the supplementary tables file (Supplementary table 1). Additionally we have now provided the complete meta-analysis summary statistic data (23,096 samples) for all SNPs from our LTL GWAS study as a supplementary file. We believe this would allow easy access for the scientific community to evaluate specific genetic associations with LTL and would facilitate subsequent larger-scale genetic analysis for leukocyte telomere length to be performed.

Reviewers' comments:

Reviewer #1 (Remarks to the Author):

The authors have adequately addressed the majority of my comments. Just a few additional comments below:

--Co-localization refers to a specific type of statistical method that quantifies the probability that two association signals are due to the same causal variant. So it does not seem correct to refer "colocalization" of a GWAS signal and a QTL signal and to provide the r^2 (LD) between lead SNPs as evidence of that co-localization. It would be better to apply a method such as coloc to your data and then provide posterior probabilities of a common causal variant.

--Can colocalization of GWAS hits with hQTLs really implicate specific genes (especially when the SNP is not an eQTL)? Seems like an hQTL could be related to the regulation of multiple genes near the histone(s) that are affected by the SNPs. If I'm wrong here, perhaps the reader needs to know more about the positions and/or functions of the histones being modified by SNPs.

Reviewer #2 (Remarks to the Author):

In my view the authors have done an excellent job in addressing the reviewer comments, provided a thorough written response and updated the manuscript materials accordingly. I have no further comments at this time.

We thank the reviewers and the editor for re-examining our manuscript and for the valuable feedback. We have provided our responses to all points raised by the reviewer below and have indicated on changes made to the revised manuscript.

Reviewer #1 (Remarks to the Author):

The authors have adequately addressed the majority of my comments. Just a few additional comments below:

--Co-localization refers to a specific type of statistical method that quantifies the probability that two association signals are due to the same causal variant. So it does not seem correct to refer "colocalization" of a GWAS signal and a QTL signal and to provide the r^2 (LD) between lead SNPs as evidence of that co-localization. It would be better to apply a method such as coloc to your data and then provide posterior probabilities of a common causal variant.

We thank the reviewer for highlighting this point. We have now rephrased the results section by replacing "co-localization" with "in LD", to firstly indicate that the 4 QTL SNPs were in strong LD with 4 GWAS lead SNPs. The manuscript text has been modified to:

"Four LTL GWAS lead SNPs from the study were in LD with four QTL SNPs (r^2 between 0.985 – 0.916, 1000G ASN panel). ... The rs7253490 GWAS lead SNP was in LD with an eQTL SNP (rs17554725, r^2 = 0.944, 1000G ASN panel) that affects *ZNF257* gene expression in both monocytes and T cells.... The rs227080 GWAS lead SNP was in LD with another eQTL SNP (rs660429, r^2 = 0.916, 1000G ASN panel) that affects *ATM* expression in T cells." (Page 9, revised manuscript).

As specifically suggested by the reviewer, we have now performed the additional co-localization analysis. For each QTL signal, we extracted summary QTL association results for all regional QTL SNPs from the BLUPRINT database (<http://blueprint-dev.bioinfo.cnio.es/WP10/>) and their corresponding LTL associations from the SCHS GWAS meta-analysis. Subsequently, evidence for a shared common genetic causal variant at each region for both QTL associations and LTL associations were assessed with the coloc.abf function in R. For the QTLs reported in the manuscript we observed moderate to high evidence for a common causal variant with the LTL GWAS signals (posterior probability between 61.71% and 96.32%). These data have been indicated in the results section on page 9 and in supplementary table 7 of the revised manuscript. We have included the following text in the revised manuscript:

"Co-localization analyses at these regions indicated moderate to high probability for shared causal variant between these QTL association signals and LTL GWAS association signals [posterior probability between 61.71% and 96.32% (Supplementary table 7)]." (Page 9, revised manuscript)

We have also indicated the methods for the coloc analysis in the methods section of the revised manuscript on page 23.

"For identified QTLs that were in LD with GWAS lead SNPs ($r^2 > 0.60$, 1000 genomes ASN panel) co-localization analysis was performed to determine the probability of a shared common causal variant for both QTL and LTL signals [87]. Analysis was performed by extracting summary association results for all

regional QTL SNPs ($P_{\text{Adj}} < 5 \times 10^{-8}$, 2Mb region) from the BLUEPRINT database (<http://blueprint-dev.bioinfo.cnio.es/WP10/>) [39] and their corresponding LTL GWAS association results from the SCHS meta-analysis and evaluated using the coloc.abf R package.” (Page 23, revised manuscript)

--Can colocalization of GWAS hits with hQTLs really implicate specific genes (especially when the SNP is not an eQTL)? Seems like an hQTL could be related to the regulation of multiple genes near the histone(s) that are affected by the SNPs. If I'm wrong here, perhaps the reader needs to know more about the positions and/or functions of the histones being modified by SNPs.

We fully agree with the reviewer that it would be challenging to draw firm conclusions on potentially implicated effector genes with hQTL data alone. We have therefore toned down the discussion section on this point in the revised manuscript. We discussed the observation that the histone peaks identified in the BLUEPRINT epigenomics database showed evidence of overlap with the genomic regions coding for *MPHOSPH6* and *POT1*. We have modified the manuscript text in the following manner:

“Two other identified histone modification peaks showed evidence of overlap with the *MPHOSPH6* and *POT1* gene regions. Whether or not the two histone peaks directly result in regulation of *MPHOSPH6* and *POT1* would require further biological investigations.” (page 13, revised manuscript)

To be more helpful to the readers, we have now extracted and included the start and end positions for histone marks in addition to the overlapped genes at these regions and the summary statistics of the QTL effects that were provided by the BLUEPRINT epigenomics study in supplementary table 7.

Reviewer #2 (Remarks to the Author):

In my view the authors have done an excellent job in addressing the reviewer comments, provided a thorough written response and updated the manuscript materials accordingly. I have no further comments at this time.

We thank the reviewer for the positive comments.

Editor's comments:

Please be aware that for certain types of new data, including most types of genetic data, journal policy is that deposition in a community-endorsed, public repository is generally mandatory prior to publication. Data submission can be a lengthy process, and we strongly suggest that you begin this well in advance of potential publication to avoid delays later on. Please include a statement about data availability (specifically GWAS summary statistics) in your point-by-point letter accompanying your revisions.

We have now indicated that the full GWAS meta-analysis data used in the study is available as part of the supplementary information in page 26 of the revised manuscript.

“Data Availability

Full GWAS meta-analysis data for all SNPs evaluated in the study is available as a Supplementary file.” (page 26, revised manuscript).

REVIEWERS' COMMENTS:

Reviewer #1 (Remarks to the Author):

The authors have addressed my comments. My final recommendation regarding colocalization analysis is to include the values for all three priors that were used. These may have been left as the default values, but these should be mentioned in the paper as the results are quite sensitive to these priors.

REVIEWERS' COMMENTS:

Reviewer #1 (Remarks to the Author):

The authors have addressed my comments. My final recommendation regarding colocalization analysis is to include the values for all three priors that were used. These may have been left as the default values, but these should be mentioned in the paper as the results are quite sensitive to these priors.

We thank the reviewer for re-evaluating our manuscript. The colocalization analysis were performed using default values and as recommended we have now indicated the values of the priors used in the methods section of the revised manuscript on page 24.

“...evaluated using the coloc.abf R package, using default priors [prior probability that a SNP is associated with first trait, $P_1 = 1 \times 10^{-4}$, prior probability that a SNP is associated with second trait, $P_2 = 1 \times 10^{-4}$, and prior probability that a trait is associated with both traits, $P_{1\&2} = 1 \times 10^{-5}$].” (page 24, revised manuscript)